# Spatiotemporal recruitment of the ubiquitin-specific protease USP8 directs endosome maturation

**Yue Miao[1,2†], Yongtao Du[1,2†], Baolei Wang[1,2], Jingjing Liang[1], Yu Liang[1,2], Song Dang[1], Jiahao Liu[2,3], Dong Li[2,3], Kangmin He[1,2]\*, Mei Ding[1,2]\***

[1]State Key Laboratory of Molecular Developmental Biology, Institute of Genetics and Developmental Biology, Chinese Academy of Sciences, Beijing, China; [2]University of Chinese Academy of Sciences, Beijing, China; [3]National Laboratory of Biomacromolecules, CAS Center for Excellence in Biomacromolecules, Institute of Biophysics, Chinese Academy of Sciences, Beijing, China

*For correspondence:
kmhe@genetics.ac.cn (KH);
mding@genetics.ac.cn (MD)

†These authors contributed equally to this work

Competing interest: The authors declare that no competing interests exist.

## eLife Assessment

The manuscript presents an **important** model for the field of endosome maturation, providing perspective on the role of the deubiquitinating enzyme UPS-50/USP8 in the process. The evidence presented in the paper is clear, incorporating well-designed experiments that suggest the dual actions of UPS-50 and USP8 in the conversion of early endosomes into late endosomes. Overall, the work is **convincing** and centers on an intriguing subject.

**Abstract** The spatiotemporal transition of small GTPase Rab5 to Rab7 is crucial for early-to-late endosome maturation, yet the precise mechanism governing Rab5-to-Rab7 switching remains elusive. USP8, a ubiquitin-specific protease, plays a prominent role in the endosomal sorting of a wide range of transmembrane receptors and is a promising target in cancer therapy. Here, we identified that USP8 is recruited to Rab5-positive carriers by Rabex5, a guanine nucleotide exchange factor (GEF) for Rab5. The recruitment of USP8 dissociates Rabex5 from early endosomes (EEs) and meanwhile promotes the recruitment of the Rab7 GEF SAND-1/Mon1. In USP8-deficient cells, the level of active Rab5 is increased, while the Rab7 signal is decreased. As a result, enlarged EEs with abundant intraluminal vesicles accumulate and digestive lysosomes are rudimentary. Together, our results reveal an important and unexpected role of a deubiquitinating enzyme in endosome maturation.

## Introduction

Endosomes are dynamic and heterogeneous organelles that act as hubs for endocytic trafficking, recycling, and degradation. During endocytosis, membrane proteins marked by ubiquitination are incorporated into endocytic vesicles and then transported to early endosomes (EEs). At EEs, membrane proteins are either brought to the recycling endosomes (REs) or incorporated into intraluminal vesicles (ILVs) with the help of ESCRT (Endosomal Sorting Complex Required for Transport) complexes (*Sardana and Emr, 2021*). The ILV-containing compartments, often referred to as multivesicular bodies (MVBs), are more acidified and upon fusion with lysosomes, can deliver both transmembrane cargoes for degradation and fresh supplies of lysosomal hydrolase enzymes required for the turnover of proteins, lipids, and other cellular components (*Huotari and Helenius, 2011*; *Sardana and Emr, 2021*). The EE-to-MVB endosome maturation process is mainly controlled by endosome-specific Rab

GTPases and phosphoinositides (*Borchers et al., 2021*; *Huotari and Helenius, 2011*; *Kümmel et al., 2023*; *Stenmark, 2009*). Rab GTPases are guanine nucleotide-binding proteins that switch between an inactive GDP-bound and an active GTP-bound state. Within the cytosol, the GDP-bound Rab proteins are kept soluble by binding to the GDP dissociation inhibitor. The GTP-bound Rabs, activated by their corresponding GEF, can associate with cellular membranes and recruit effector proteins.

Rab5, the Rab GTPase critical for EE formation and function, can be activated and recruited to EEs by its GEF, Rabex5 (*Mattera and Bonifacino, 2008*; *Zhu et al., 2007*). The GTP-bound active Rab5 recruits its effectors, for instance, Rabaptin-5. Rabaptin-5 in turn binds to Rabex5 via a coiled-coil region, resulting in a positive feedback loop of Rab5 activation on endosomal membranes (*Zhang et al., 2014b*). Isolated Rabex5 appears to have relatively low GEF activity in vitro (*Delprato and Lambright, 2007*) and binding of Rabaptin-5 to Rabex5 causes a rearrangement in the Rabex5 C-terminus and subsequently enhances nucleotide exchange of Rab5 (*Horiuchi et al., 1997*; *Lauer et al., 2019*; *Lippé et al., 2001*; *Zhang et al., 2014b*). Thus, autoinhibition likely serves as a key regulatory mechanism in controlling GEF activity. Other effectors of Rab5 include the phosphatidylinositol-3-phosphate (PI3P) kinase complex II (VPS34/VPS15/Beclin 1/UVRAG) (*Tremel et al., 2021*), which promotes the synthesis of PI3P on endosomes, and the class C core vacuole/endosome tethering (CORVET) complex, which promotes membrane homotypic fusion (*Abenza et al., 2010*; *Balderhaar et al., 2013*; *Peplowska et al., 2007*). Rab5 also recruits the early endosomal antigen 1 (EEA1), a tethering protein required for fusion of endocytic vesicles with EEs (*Christoforidis et al., 1999*). The phosphorylated head group of PI3P binds to specific domains, such as the FYVE domain in EEA1, allowing for the organelle-specific association of proteins containing this domain (*Lawe et al., 2002*; *Stenmark et al., 1996*). The fusion of vesicles with organelles depends on tethering complexes as well as soluble *N*-ethylmaleimide-sensitive factor attachment protein receptors (SNAREs). Reconstitution of EE fusion revealed that EEA1 functions in the context of multiple additional components, including endosomal SNAREs, during fusion (*Ohya et al., 2009*). Rab5 effectors also include the Rab7 GEF Mon1-Ccz1 complex. Rab7 activation by Mon1-Ccz1 complex is essential for the biogenesis and positioning of late endosomes (LEs) and lysosomes, and for the fusion of endosomes and autophagosomes with lysosomes. The Mon1-Ccz1 complex is a heterodimer in yeast, and a heterotrimer in metazoan cells (*Dehnen et al., 2020*; *Vaites et al., 2018*; *van den Boomen et al., 2020*; *Wang et al., 2002*). The activation of Mon1-Ccz1 is depended on Rab5 interaction (*Borchers et al., 2023*; *Langemeyer et al., 2020*) and Mon1 also coordinates endosome maturation together with the Rab5 GAP TBC1D18 (*Hiragi et al., 2022*). Additionally, the Mon1-Ccz1 complex is able to interact with Rabex5 (*Poteryaev et al., 2010*), causing dissociation of Rabex5 from the membrane, which probably terminates the positive feedback loop of Rab5 activation and then promotes the recruitment and activation of Rab7 on endosomes (*Nordmann et al., 2010*; *Poteryaev et al., 2010*). All in all, the hierarchy of protein recruitment and their reciprocal regulation ensures the rapid transition from Rab5- to Rab7-positive vesicles which is critical for endosome maturation.

The conserved ESCRT machinery drives endosomal membrane deformation and scission leading to the formation of ILVs within MVBs. On the surface of endosomes, ESCRT-0, -I, and -II bind to ubiquitinated membrane proteins, while ESCRT-III and Vps4 bud ILVs into the lumen of the endosomes (*Wollert and Hurley, 2010*). The deubiquitinating enzyme (DUB) USP8 belongs to the ubiquitin-specific protease (USP) family and plays a prominent role in the regulation of endosomal sorting of a wide range of transmembrane receptors (*Dufner and Knobeloch, 2019*). In addition to direct deubiquitination, USP8 also regulates protein stability through its interactions with various ESCRT components. By binding to STAM and Hrs, USP8 stabilizes the ESCRT-0 complex, which is the crucial element directing ubiquitinated substrates toward MVB/lysosome-mediated degradation (*Mizuno et al., 2006*; *Row et al., 2006*). The N-terminal microtubule interacting and transport domain of USP8 interacts with ESCRT-III proteins, which are thought to be the driving force of several membrane scission events (*Row et al., 2007*). In the absence of USP8, EGFR is abnormally accumulated in EEs (*Alwan and van Leeuwen, 2007*; *Row et al., 2006*). Additionally, the endocytic trafficking of the Frizzled receptor, Smooth, as well as key components of the Wnt and Hedgehog pathways is also affected by *usp8* deficiency (*Ali et al., 2013*; *Berlin et al., 2010a*; *Berlin et al., 2010b*; *Crespo-Yàñez et al., 2018*; *Jacomin et al., 2015*; *MacDonald et al., 2014*). Noticeably, however, the impact of USP8 on protein stability is highly variable and sometimes controversial (*Berlin et al., 2010b*; *Mizuno et al., 2005*; *Niendorf et al., 2007*; *Row et al., 2006*).

Here, utilizing both *Caenorhabditis elegans* and cultured mammalian cells as models, we identified that USP8 is recruited to Rab5-positive vesicles through Rabex5 and functions through both the endosomal dissociation of Rabex5 and recruitment of SAND-1/Mon1 to promote endosome maturation. USP8 has recently arisen as a promising therapeutic target in at least two distinct pathological contexts associated with USP8 overexpression or gain-of-function variants (*Dufner and Knobeloch, 2019*; *Li et al., 2024*; *Reincke et al., 2015*). Thus, our data reveal an important and direct role of USP8 in endosome maturation and have clear implications for the therapeutic treatment of USP8-related human diseases.

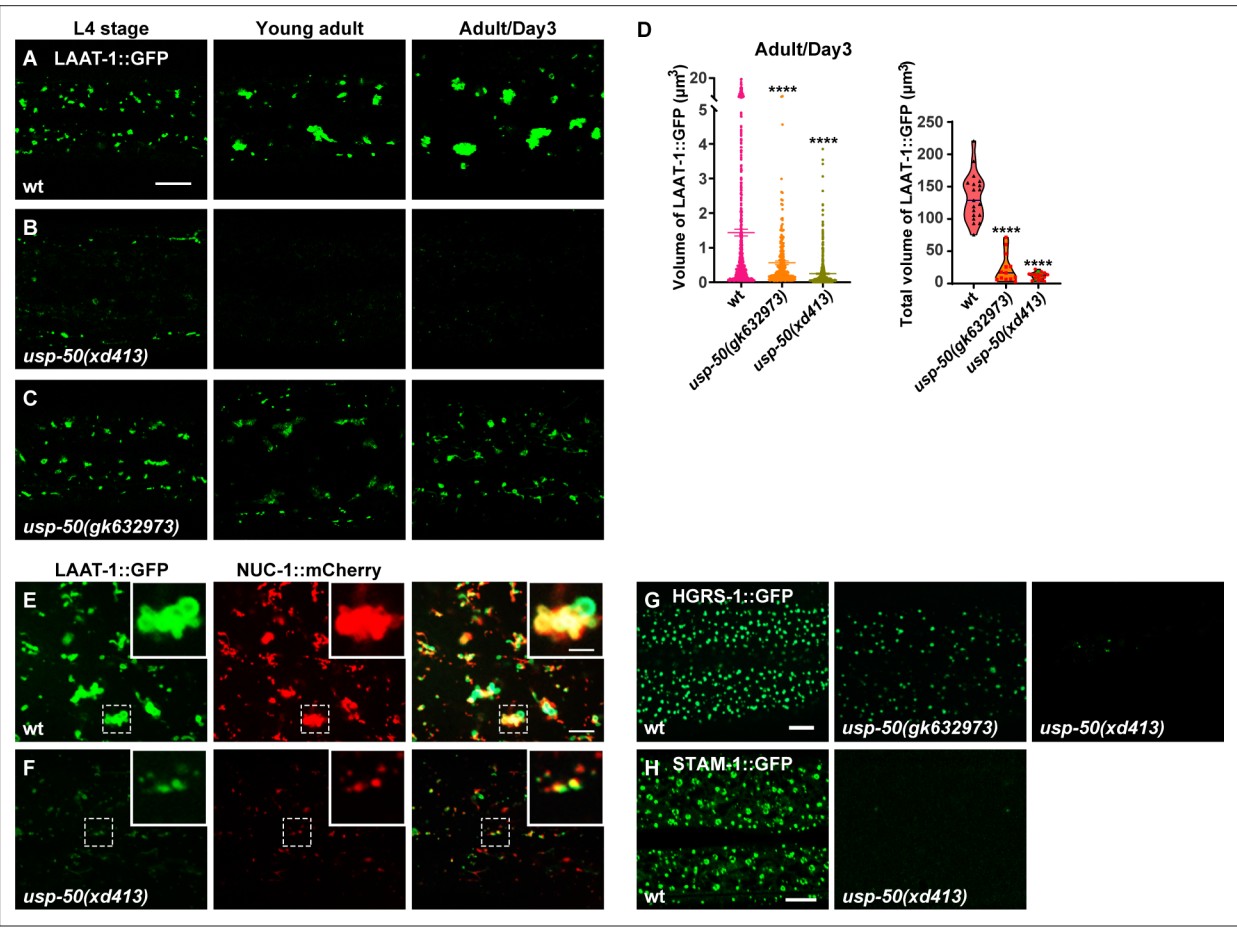

**Figure 1.** Abnormal lysosome morphology and enlarged multivesicular body (MVB)-like structures in *usp-50* mutants. (**A–C**) Confocal fluorescence images of hypodermal cell 7 (hyp7) expressing the LAAT-1::GFP marker to highlight lysosome structures in L4 stage, young adult, and 3-day-old adult animals. Scale bar: 5 µm. (**D**) Quantification of the individual volume and total volume of LAAT-1::GFP vesicles in hyp7 of 3-day-old adults. 19 animals for wild-type, 16 animals for *usp-50(xd413)*, and 13 animals for *usp-50(gk632973)* were quantified. ****p<0.0001. One-way ANOVA with Tukey's test. (**E, F**) The lysosomal membrane marker LAAT-1::GFP co-localizes with the lysosomal hydrolase NUC-1::mCherry in both wild-type (**E**) and *usp-50(xd413)* (**F**) animals. Scale bar represents 5 µm for (**E, F**) and 2 µm for enlarged inserts. (**G**) Confocal fluorescence images of hypodermis expressing HGRS-1::GFP in L4 stage worms. Compared to wild-type, HGRS-1::GFP signal is reduced in *usp-50(gk632973)* and *usp-50(xd413)* animals. Scale bar: 5 µm. (**H**) Confocal fluorescence images of hypodermis expressing STAM-1::GFP in L4 stage worms. Compared to wild-type, STAM-1::GFP signal is reduced in *usp-50(xd413)* animals. Scale bar: 5 µm.

The online version of this article includes the following source data and figure supplement(s) for figure 1:

**Source data 1.** Excel file containing the quantified data of statistic analysis for *Figure 1D*.

**Figure supplement 1.** ESCRT-0 components are stabilized by *usp-50*.

**Figure supplement 1—source data 1.** Excel file containing the quantified data of statistic analysis for *Figure 1—figure supplement 1A and B*.

## Results

### *usp-50* mutation leads to reduction of lysosomal compartments

The epithelial cell hyp7 (hypodermal cell 7) covers most of the worm body and is the largest cell in *C. elegans*. When fused to GFP and driven by the *ced-1* promoter, the transmembrane lysine/arginine transporter LAAT-1 can be used to highlight the lysosome structures in the hyp7 cell (*Figure 1A*; *Liu et al., 2012*). In wild-type, as worms mature from L4 (larval stage 4) to adult stage, the vesicular and tubular structures of lysosomes become enlarged (*Figure 1A*; *Sun et al., 2020*). In *usp-50(xd413)*, a recessive mutation isolated from a genetic screen of worms mutagenized with EMS (ethylmethane sulfonate), lysosomes appear to be greatly reduced in size (*Figure 1B and D*). For another *usp-50* allele, *gk632973*, the lysosome distribution and morphology appear normal during L4 stage (*Figure 1C*). However, when *usp-50(gk632973)* animals reach adult stage, the individual size and total volume of lysosomes are also reduced (*Figure 1C and D*). *nuc-1* encodes a lysosomal deoxyribonuclease (*Wu et al., 2000*). In *usp-50(xd413)* animals, the NUC-1 protein is properly targeted to LAAT-1-containing vesicles (*Figure 1E and F* and *Figure 1—figure supplement 1A*), which indicates that the assembly of de novo lysosomes is not affected by *usp-50*.

usp-50 encodes the *C. elegans* homolog of *Saccharomyces cerevisiae* Doa4p and human USP8/UBPY (*Bowers et al., 2004*; *Huynh et al., 2016*). All the USP8 family members contain a ubiquitin C-terminal hydrolase (UCH) domain which is crucial for the deubiquitination activity (*Komander et al., 2009*). The molecular lesions of both *usp-50(xd413)* and *usp-50(gk632973)* occur in the UCH domain, leading to a G541A and A523V substitution, respectively.

USP8 was identified as a protein associated with ESCRT components (*Kato et al., 2000*; *Mathieu et al., 2022*; *Row et al., 2007*). The ESCRT machinery drives endosomal membrane deformation and scission, leading to the formation of ILVs within MVBs (*Babst, 2011*; *Wollert and Hurley, 2010*). In wild-type animals, the two ESCRT-0 components, HGRS-1 and STAM-1, are distributed in a distinct punctate pattern (*Figure 1G and H*). When *usp-50* is mutated, the punctate HGRS-1::GFP and STAM-1 signals are greatly reduced (*Figure 1G and H* and *Figure 1—figure supplement 1B*), which is consistent with the role of USP8 in stabilizing the ESCRT-0 complex (*Kato et al., 2000*; *Mizuno et al., 2006*; *Niendorf et al., 2007*; *Row et al., 2006*; *Zhang et al., 2014a*).

### Enlarged EEs in *usp-50/usp8* mutant cells

Lysosomes receive and digest materials generated by endocytic pathways. Using YFP-tagged 2xFYVE (YFP::2xFYVE), which specifically labels PI3P, we examined the size and morphology of EEs (*Figure 2A*). We found that the size of individual EEs is significantly increased (*Figure 2B*). Meanwhile, the total volume of EEs in *usp-50* mutants remains similar to wild-type (*Figure 2B*), implying that the homotypic fusion of EEs is probably increased. The enlarged EEs in *usp-50(xd413)* are not coated with the lysosome marker LAAT-1::mCherry (*Figure 2C and D* and *Figure 2—figure supplement 1A*). Hence, the EEs remain differentiated from lysosome structures. The enlarged EE defect in *usp-50(xd413)* mutants is rescued by either worm *usp-50* or human USP8 (*Figure 2G and J* and *Figure 2—figure supplement 1B*), which suggests that the function of USP-50/USP8 is evolutionarily conserved. Indeed, when we knocked out the expression of USP8 (USP8-KO) in human breast cancer SUM159 cells (*Figure 2—figure supplement 1C*), the transient expression of EGFP-2xFYVE confirmed that USP8 depletion caused EE enlargement (*Figure 2K*).

Next, we prepared high-pressure frozen samples and performed transmission electron microscopy (TEM) analysis to examine worm epidermal cells. Compared to wild-type, many abnormal large vesicles with various intraluminal contents were detected in *usp-50(xd413)* mutants (*Figure 2M–Q*). Based upon the vesicular inclusions, we divided those vesicles into four categories. The first class is composed of vesicles containing abundant intraluminal vesicle structures (*Figure 2N*). About 58.7% of the abnormal vesicles belong to this category. The second class (26.8%) includes vesicles filled with threadlike membrane structures (*Figure 2Q*). Vesicles of the third type (22.4%) are loaded with threadlike membrane structures and electron-dense material (*Figure 2P*). The fourth class, which accounts for a relatively small proportion of mutant vesicles (4.9%), contains various cellular organelles, for instance mitochondria (*Figure 2Q*). Given the significantly reduced LAAT-1-coated structures and the increased size of FYVE-positive vesicles observed in *usp-50* mutants, it is plausible that these enlarged vesicles represent abnormally expanded EEs.

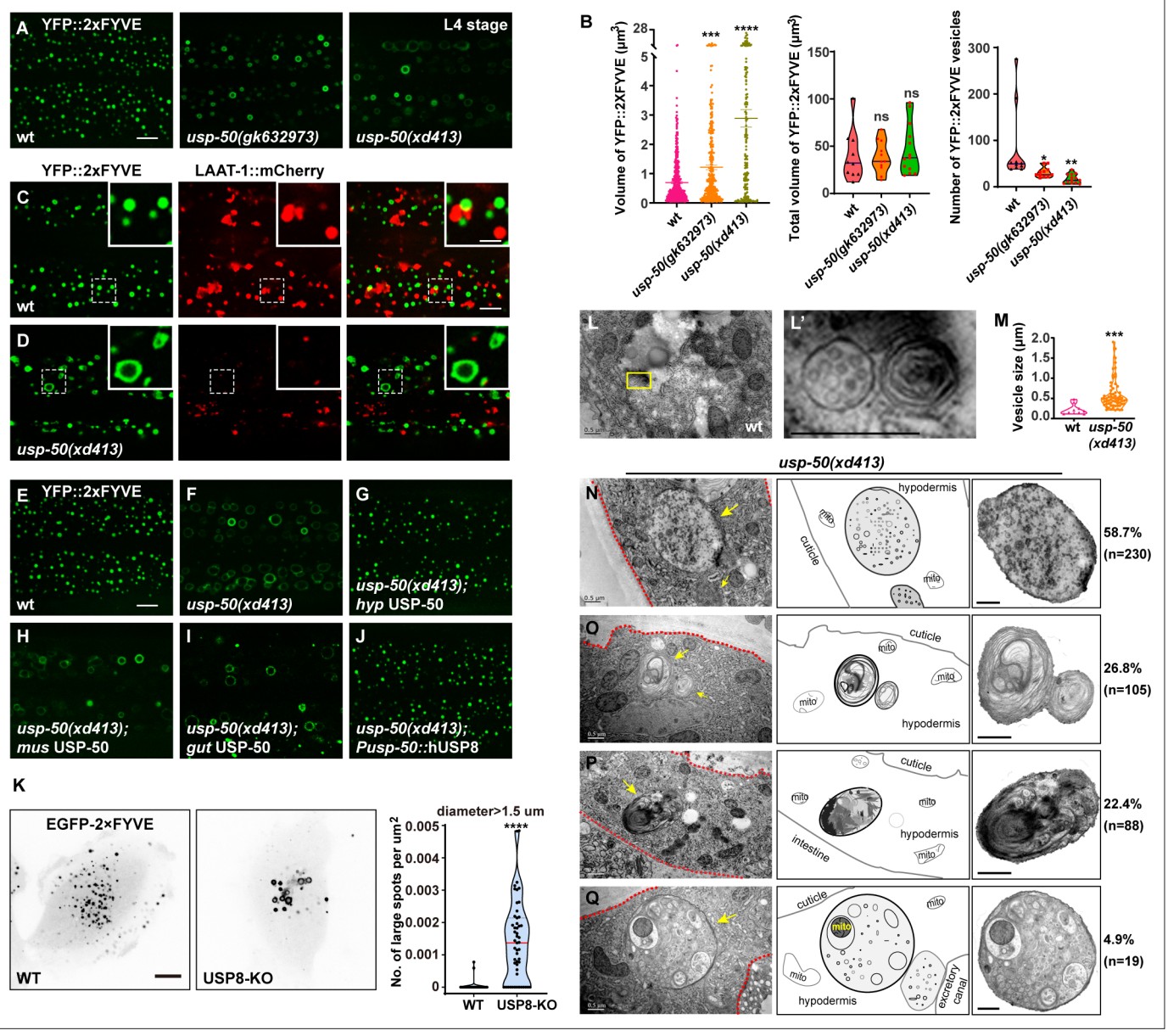

**Figure 2.** Enlarged early endosomes (EEs) in *usp-50/usp8* mutant cells. (**A**) Confocal fluorescence images of hypodermis expressing YFP::2xFYVE to detect EEs in L4 stage animals. Compared to wild-type, EEs are enlarged in *usp-50(gk632973)* and *usp-50(xd413)* mutants. Scale bar: 5 μm. (**B**) Quantification of the individual vesicle size, total volume, and number of YFP::2xFYVE-marked vesicles in hyp7 in L4 worms (10 animals for wild-type, 11 animals for *usp-50(gk632973)*, 13 animals for *usp-50(xd413)*). Data are presented as mean ± SEM. *p<0.05. **p<0.01. ***p<0.001. ****p<0.0001. ns, not significant; one-way ANOVA with Tukey's test. (**C, D**) The YFP::2xFYVE marker is not co-localized with LAAT-1::mCherry in wild-type (**C**) or *usp-50(xd413)* animals (**D**). Scale bar represents 5 μm for (**C, D**) and 2 μm for enlarged inserts. (**E–J**) The YFP::2xFYVE pattern in wild-type (**E**), *usp-50(xd413)* mutants (**F**), *usp-50(xd413)* with hypodermis-specific USP-50 expression (**G**), *usp-50(xd413)* with muscle-specific USP-50 expression (**H**), *usp-50(xd413)* with gut-specific USP-50 expression (**I**), and *usp-50(xd413)* with expression of human USP8 driven by the *usp-50* promoter (**J**). Scale bar: 5 μm. (**K**) Distribution of transiently expressed EGFP-2xFYVE in wild-type and USP8-KO SUM159 cells. Both wild-type and USP8-KO SUM159 cells were transiently transfected with EGFP-2xFYVE and then imaged by spinning-disk confocal microscopy. The right panel shows quantification of the number of large endosomes marked by EGFP-2xFYVE from 38 cells for wild-type and 51 cells for USP8-KO (****p<0.0001; unpaired Student's t-test). Scale bar: 10 μm. (**L, L'**) High-pressure freezing EM reveals the multivesicular body (MVB) structures in wild-type animals. The yellow boxed area is enlarged in (**L'**). (**M**) Quantification of the abnormal enlarged vesicle in *usp-50(xd413)* animals. For comparative analysis, the quantification of MVB structures in wild-type animals was also included. Data are presented as mean ± SEM. ***p<0.001. Unpaired Student's t-test was performed. (**N–Q**) The enlarged abnormal vesicles in *usp-50(xd413)*, including vesicles containing abundant intraluminal vesicles (58.7%) (**N**), vesicles filled with threadlike membrane structures (26.8%) (**O**), vesicles loaded with electron-dense material (22.4%) (**P**), and vesicles containing various cellular organelles (4.9%) (**Q**). Red dashed lines indicate hypodermal cells. Yellow arrows indicate representative vesicles. mito, mitochondrion. Scale bar represents 0.5 μm for (**L–Q**).

*Figure 2 continued on next page*

*Figure 2 continued*

The online version of this article includes the following source data and figure supplement(s) for figure 2:

**Source data 1.** Excel file containing the quantified data of statistic analysis for *Figure 2B, K, and M*.

**Figure supplement 1.** *usp-50* functions cell-autonomously.

**Figure supplement 1—source data 1.** Original file for the western blot analysis in *Figure 2—figure supplement 1C* (anti-USP8 and anti-GAPDH).

**Figure supplement 1—source data 2.** PDF containing original scans of the relevant western blot analysis (anti-USP8 and anti-GAPDH) with highlighted bands and sample labels for *Figure 2—figure supplement 1C*.

**Figure supplement 1—source data 3.** Excel file containing the quantified data of statistic analysis for *Figure 2—figure supplement 1A and B*.

**Figure supplement 2.** Mutation of *usp-50* does not affect other organelles.

There was no obvious alteration in the pattern of reporters for Golgi apparatus (MANS::GFP), retromers (VPS-29::GFP), or REs (GFP::RME-1) (*Figure 2—figure supplement 2*). This suggests that the endolysosomal trafficking process is rather specifically affected by *usp-50*.

## USP8 is dynamically recruited to Rab5-positive vesicles

How does USP-50/USP8 control the size of EEs? To address this question, we firstly analyzed the subcellular localization of USP-50/USP8. In worm epidermal cells, the GFP-tagged USP-50 protein is co-localized with mCherry::RAB-5 (*Figure 3—figure supplement 1A*) but not with mCherry::R-AB-7 (*Figure 3—figure supplement 1B*), consistent with the early endosomal localization of USP-50/USP8 (*Mizuno et al., 2005*; *Row et al., 2006*). The catalytic UCH domain possesses a 'cysteine box' containing the active site residues, including Cysteine 492 (C492) (*Naviglio et al., 1998*). When we mutated C492 to Alanine, we found that the GFP-tagged USP-50(C492A) protein is still co-localized with mCherry::RAB-5 (*Figure 3—figure supplement 1C*). Thus, the EE localization of USP-50 is not dependent on its deubiquitination activity. Meanwhile, the mCherry::RAB-5 signal is strongly enhanced by the overexpression of USP-50(C492A) (*Figure 3—figure supplement 1C*), suggesting that the C492A mutation may have a dominant-negative effect on USP-50 function.

To reveal the subcellular localization of USP8 in mammalian cells, we used the CRISPR/Cas9 approach to generate a SUM159 cell line in which the endogenous USP8 was tagged with mEGFP (USP8-mEGFP$^{+/+}$) (*Figure 3—figure supplement 2*). As shown in *Figure 3*, USP8-mEGFP co-localizes well with the EE marker mScarlet-I-EEA1 (*Figure 3A*). In contrast, LE or lysosome markers, including mScarlet-I-Rab7a and Lamp1-mScarlet-I, have little overlap with the USP8-mEGFP signal (*Figure 3B and C*). Constitutively activated Rab5 GTPase (Rab5c-Q80L) causes enlarged EEs (*Stenmark et al., 1994*; *Wegner et al., 2010*). Instead of being evenly distributed, USP8-mEGFP formed individual spots located on one side of the enlarged endosomes (*Figure 3D*). We further analyzed the sub-organelle distribution of USP8 using structured illumination microscopy (SIM). Consistent with what we observed in enlarged EEs caused by Rab5c-Q80L overexpression, the USP8-mEGFP dots indeed located at subdomains of EEs (*Figure 3E*). The uneven distribution of USP8 implies that USP8 may associate dynamically with EEs. To further investigate the dynamic association of USP8 with EEs, we transiently expressed mScarlet-I-Rab5c in the genome-edited USP8-mEGFP$^{+/+}$ cells and tracked the recruitment dynamics of USP8-mEGFP to EEs by spinning-disk confocal microscopy (*Figure 3F*). We previously reported that Rab5 was recruited to nascent endocytic carriers following the uncoating of clathrin-coated vesicles (*He et al., 2017*). This recruitment resulted in the creation of Rab5-positive endocytic carriers, which could subsequently fuse with other Rab5-positive endocytic carriers or EEA1-positive EEs (*He et al., 2017*). We found that USP8-mEGFP was recruited to these nascent mScarlet-I-Rab5c-positive vesicles which appeared around the bottom surface of the cells (*Figure 3F*). Additionally, we observed the dynamic appearance of single USP8-mEGFP spots on these large Rab5c-positive endosomes (*Figure 3G and H*).

## USP-50/USP8 recruitment dissociates RABX-5 from endosomes

How is the endosomal recruitment of USP8 related to its function in endolysosomal trafficking process? By searching the putative USP-50-binding partners, we found that USP-50 can bind to RABX-5, the worm homolog of Rabex5 protein (*Figure 4A and B*). To determine which domain of RABX-5 is required for the interaction with USP-50, we constructed a panel of FLAG-tagged RABX-5 truncation mutants (*Figure 4C*) and performed a series of immunoprecipitation tests (*Figure 4D*). In general, the

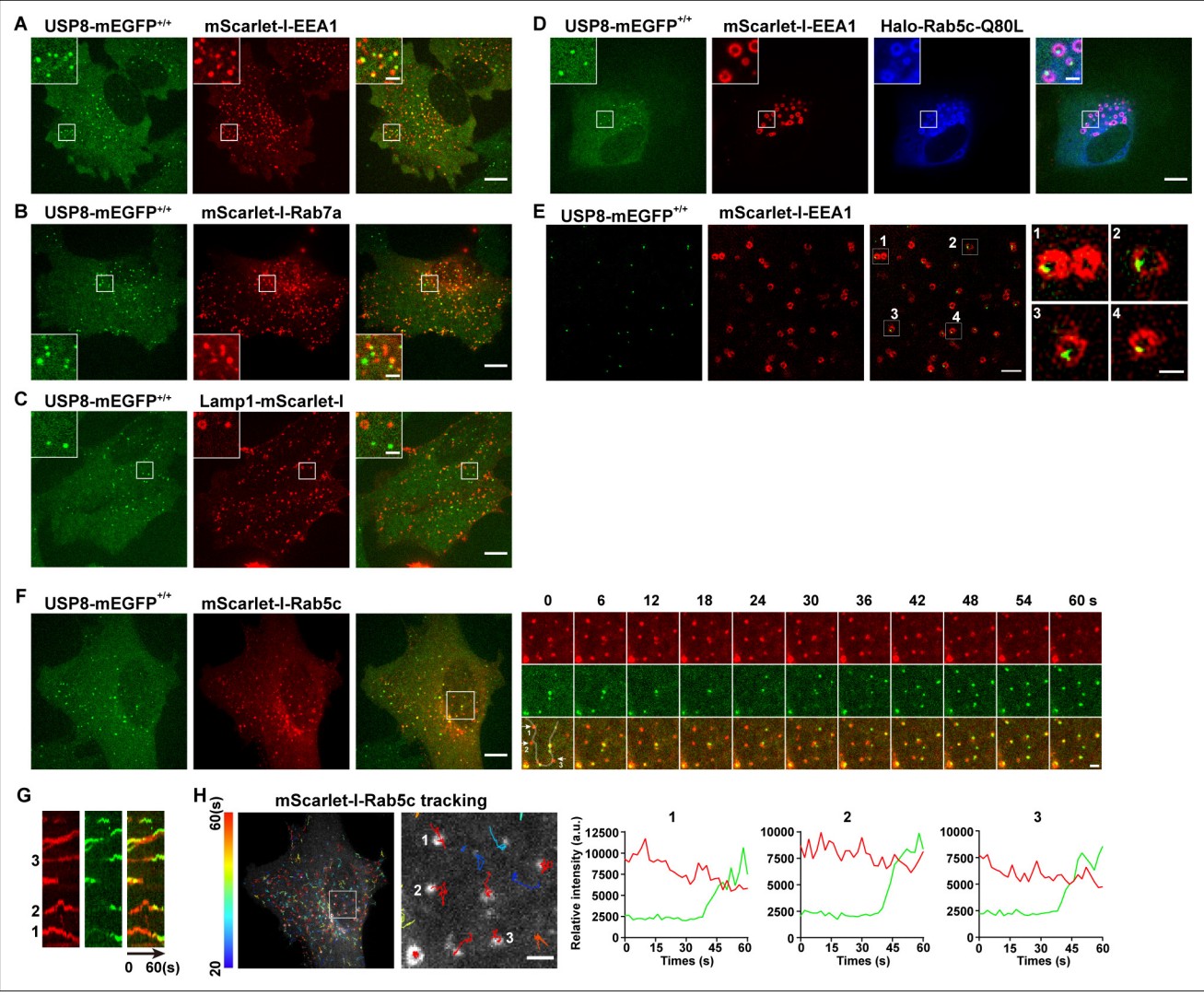

**Figure 3.** USP8 is recruited to Rab5-positive vesicles. (**A–C**) SUM159 cells genome-edited for USP8-mEGFP⁺/⁺ were transiently transfected with the indicated mScarlet-I-tagged proteins and then imaged by spinning-disk confocal microscopy. The single-frame image in the figure shows the distribution of USP8 with EEA1, Rab7a, or Lamp1. USP8-mEGFP is co-localized with the early endosome (EE) marker mScarlet-I-EEA1 in genome-edited USP8-mEGFP⁺/⁺ SUM159 cells (**A**). USP8-mEGFP is partly co-localized with the late endosome marker mScarlet-I-Rab7a (**B**) or the lysosome marker Lamp1-mScarlet-I (**C**). Scale bar represents 10 μm for (**A–C**) and 2 μm for enlarged inserts. (**D**) USP8-mEGFP localization on enlarged EEs in the genome-edited SUM159 cells. SUM159 cells genome-edited for USP8-mEGFP⁺/⁺ were transiently transfected with mScarlet-I-EEA1 and Halo-Rab5c-Q80L, labeled with the JF₆₄₆-HaloTag ligand, and then imaged by spinning-disk confocal microscopy. Scale bar represents 10 μm for (**D**) and 2 μm for enlarged inserts. (**E**) Structured illumination microscopy (SIM) images showing the sub-organelle localization of USP8-mEGFP on EEs marked with mScarlet-I-EEA1. USP8-mEGFP⁺/⁺ cells were transiently transfected with mScarlet-I-EEA1 and then imaged near the middle plane by SIM. Scale bar represents 2 μm for the left panels and 0.5 μm for the right enlarged panel. (**F**) SUM159 cells genome-edited for USP8-mEGFP⁺/⁺ were transiently transfected with mScarlet-I-Rab5c and then imaged at two planes (starting from the bottom plane, spaced by 0.5 μm) every 2 s (for 1 min) by spinning-disk confocal microscopy. Shown is the first frame of the maximum intensity projection of the two planes. The boxed region is enlarged and the images at the indicated times are shown on the right. Scale bar represents 10 μm for the left images and 2 μm for the right enlarged images. (**G**) Kymographs along the line (width 3 pixels) on the first merged image of the montage in (**F**) showing dynamic recruitment of USP8-mEGFP to the pre-existing mScarlet-I-Rab5-positive vesicles. (**H**) The time-lapse images in (**F**) were analyzed by single-particle tracking. The trajectories (longer than 20 s, color-coded based on track lifetime) of Rab5c are plotted on the first frame of mScarlet-I-Rab5c. The boxed region is enlarged and shown on the right. The fluorescence intensity traces for mScarlet-I-Rab5c (red) and USP8-mEGFP (green) of the three tracked events are shown. Scale bar: 2 μm.

The online version of this article includes the following source data and figure supplement(s) for figure 3:

**Source data 1.** Excel file containing the quantified data of statistic analysis for *Figure 3H*.

**Figure supplement 1.** The early endosome (EE) localization of USP-50 and USP8.

**Figure supplement 2.** The generation of endogenous tagged USP8.

*Figure 3 continued on next page*

*Figure 3 continued*

**Figure supplement 2—source data 1.** Original file for the genomic PCR analysis in *Figure 3—figure supplement 2B*.

**Figure supplement 2—source data 2.** PDF containing original file for the genomic PCR analysis with highlighted bands and sample labels for *Figure 3—figure supplement 2B*.

**Figure supplement 2—source data 3.** Original file for the western blot analysis in *Figure 3—figure supplement 2C* (anti-USP8 and anti-GAPDH).

**Figure supplement 2—source data 4.** PDF containing original scans of the relevant western blot analysis (anti-USP8 and anti-GAPDH) with highlighted bands and sample labels for *Figure 3—figure supplement 2C*.

C-terminal coiled-coil (CC) and the C-terminal proline-rich (PR) domains are sufficient for RABX-5 to interact with USP-50. Meanwhile, the N-terminal region, including an A20-zinc finger domain (ZF), a motif interacting with ubiquitin (U), the membrane-binding motif (MB), and the downstream helical bundle domain (HB) of RABX-5 could also mediate USP-50 binding. Further including Vps9 domain enhances the molecular interaction between RABX-5 and USP-50 (*Figure 4C and D*).

Does RABX-5 binding affect the endosomal localization of USP-50? To address this question, we created a molecular null of *rabx-5* using the CRISPR/Cas9 technique (*Dickinson et al., 2013*; *Figure 4E*). The *rabx-5(null)* animals are healthy and fertile and do not display obvious morphological or behavioral defects. In *rabx-5(null)* mutant animals, the punctate USP-50::GFP signal becomes diffusely distributed (*Figure 4F and G*). Thus, *rabx-5* is required for the endosomal localization of USP-50. Does USP-50 regulate endosome maturation via RABX-5? In *rabx-5(null)* mutant animals, EEs labeled with 2xFYVE are indistinguishable from those in wild-type controls (*Figure 4H and I*). Furthermore, when *rabx-5(null)* was introduced into *usp-50(xd413)* mutant backgrounds, the enlarged EE phenotype of *usp-50(xd413)* was efficiently suppressed (*Figure 4H and I*). These findings suggest that USP-50 likely exerts its regulatory effects on EE size through RABX-5.

In wild-type animals, the fluorescence signal of RABX-5::GFP in hyp7 is rather dim (*Figure 5A*). When *usp-50* is mutated, the RABX-5::GFP KI signal is greatly enhanced (*Figure 5B*). USP8-KO also caused a significant increase in the number of enlarged endosomes in Rabex5-mEGFP$^{+/+}$ SUM159 cells (*Figure 5—figure supplement 1A–C and E*). Rabex5-mEGFP signals were enriched and co-localized well with the mScarlet-I-Rab5c signal on the enlarged endosomes (*Figure 5—figure supplement 1A*). Antibody staining of endogenous EEA1 further showed that the enlarged EEs in USP8-KO cells were coated with both Rabex5 and EEA1 (*Figure 5—figure supplement 1D*). The increased endosomal RABX-5/Rabex5 may lead to Rab5 signal enhancement. Indeed, in *usp-50* mutants, the GFP::RAB-5 KI-labeled vesicles are significantly enlarged (*Figure 5—figure supplement 2A–C*) and the proportion of membrane-associated GFP::RAB-5 KI is also increased (*Figure 5—figure supplement 2D*). The GTP-bound activated RAB-5 protein binds to its downstream effector EEA1 via the N-terminal domain of EEA1 (EEA1-NT). We utilized EEA1-NT (*Figure 5—figure supplement 2E*) to show that loss of *usp-50* indeed led to more activated RAB-5 in vivo (*Figure 5—figure supplement 2F*). In addition, the total RAB-5 protein level is increased by *usp-50* mutation (*Figure 5—figure supplement 2G and H*), which implies that RAB-5 activation may stabilize RAB-5 protein. To explore the relationship between Rab5 and USP8 in SUM159 cells, we knocked down the expression of USP8 by siRNA (USP8-KD) in the clonal genome-edited EGFP-Rab5c$^{+/+}$ cells. We found that EEs, labeled by Rab5, are significantly enlarged (*Figure 5—figure supplement 2I and J*). Taken together, these results indicate that USP-50/USP8 recruitment dissociates RABX-5 from endosomes, and subsequently diminish Rab5 signaling.

## USP-50 acts on K323 deubiquitination to regulate RABX-5 localization

The enzyme-inactive USP-50(C492A) cannot rescue the enhanced endosomal RABX-5 signal in *usp-50* mutant animals (*Figure 5C and D*). This suggests that USP-50 acts through its deubiquitination activity to dissociate RABX-5 from endosomes. To further dissect how the USP-50-mediated deubiquitination might contribute to RABX-5 localization, we precipitated the FLAG-tagged RABX-5 from overexpressed 293T cells and performed ubiquitination proteomics analysis. We found that the K88 and K323 residues of RABX-5 are modified by ubiquitin in vivo (*Figure 5—figure supplement 3A and B*). K88 is located in the membrane-binding motif, while K323 resides in the conserved Vps9 domain of RABX-5 (*Figure 5—figure supplement 3B*). To understand whether and how these two ubiquitin modification sites are involved in USP-50-mediated deubiquitination, we generated non-ubiquitinated mutations at K88 and K323 (K88R and K323R, respectively). In wild-type animals, the RABX-5::GFP KI

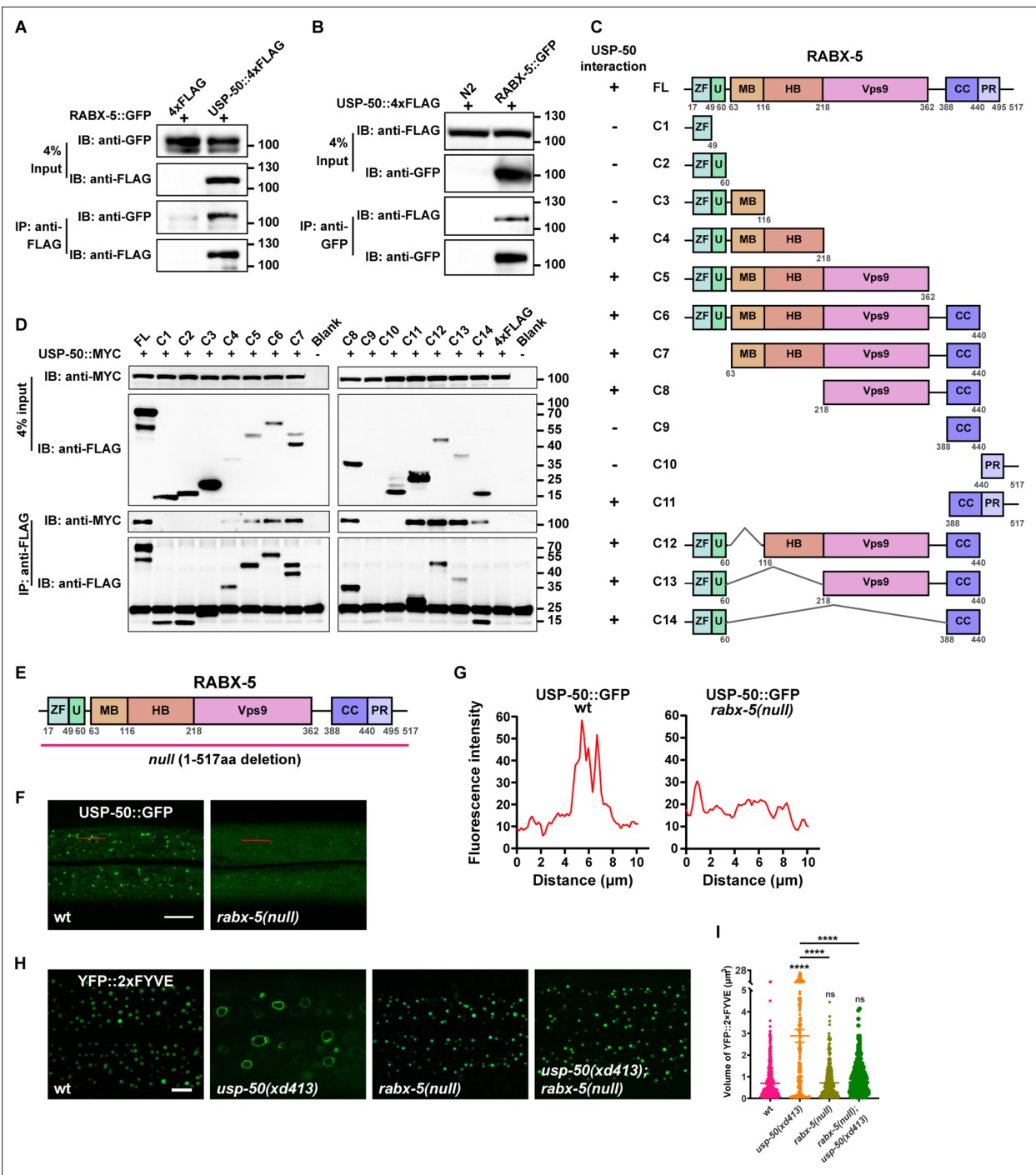

**Figure 4.** USP-50 interacts with RABX-5. (**A**) The affinity-purified RABX-5::GFP from worm lysates is immunoprecipitated by USP-50::4xFLAG purified from HEK293T cells using anti-FLAG beads. Only the area of the blot containing the USP-50::4xFLAG band is displayed. (**B**) The affinity-purified USP-50::4xFLAG from HEK293T cells is immunoprecipitated by RABX-5::GFP purified from worm lysates using anti-GFP beads. N2 is wild-type. (**C**) Schematic drawing of RABX-5 showing the domains that interact with USP-50. (**D**) Immunoprecipitation tests between USP-50::MYC and FLAG-tagged truncated RABX-5. USP-50::MYC and FLAG-tagged truncated RABX-5 were expressed via co-transfection into HEK293T cells, immunoprecipitated with FLAG beads, and immunoblotted with antibodies against MYC and FLAG. (**E**) Schematic drawing of the RABX-5 protein structure. The molecular lesion of the null mutant is indicated. (**F**) Confocal fluorescence images of hyp7 expressing USP-50::GFP in wild-type L4 animals (*wt*), *rabx-5(null)* L4 mutants. Scale bar represents 5 µm. (**G**) Line scan analyses show the fluorescence intensity values along the red solid lines in (**F**). (**H**) The early endosome (EE) (labeled by YFP::2xFYVE) enlargement phenotype of *usp-50(xd413)* could be suppressed by *rabx-5(null)* mutation. (**I**) Quantification of the volume of individual EEs in various genotypes. 10 or over animals were examined in each genotype. Data are presented as mean ± SEM. ****p<0.0001. ns, not significant. One-way ANOVA with Tukey's test.

*Figure 4 continued on next page*

*Figure 4 continued*

The online version of this article includes the following source data for figure 4:

**Source data 1.** Original file for the western blot analysis in *Figure 4A* (anti-GFP and anti-FLAG).

**Source data 2.** PDF containing original scans of the relevant western blot analysis (anti-GFP and anti-FLAG) with highlighted bands and sample labels for *Figure 4A*.

**Source data 3.** Original file for the western blot analysis in *Figure 4B* (anti-GFP and anti-FLAG).

**Source data 4.** PDF containing original scans of the relevant western blot analysis (anti-GFP and anti-FLAG) with highlighted bands and sample labels for *Figure 4B*.

**Source data 5.** Original file for the western blot analysis in *Figure 4D* (anti-MYC and anti-FLAG).

**Source data 6.** PDF containing original scans of the relevant western blot analysis (anti-MYC and anti-FLAG) with highlighted bands and sample labels for *Figure 4D*.

**Source data 7.** Excel file containing the quantified data of statistic analysis for *Figure 4G and I*.

intensity is rather dim (*Figure 5A*). When *usp-50* is mutated, the RABX-5::GFP KI intensity is strongly enhanced and displays the typical punctate pattern (*Figure 5B and I*). On a wild-type background, both RABX-5(K88R)::GFP KI and RABX-5(K323R)::GFP KI display weak signals (*Figure 5E and G*) similar to wild-type RABX-5::GFP KI (*Figure 5A*). When *usp-50* is mutated, the RABX-5(K88R)::GFP KI signal is greatly enhanced and displays an apparent punctate pattern (*Figure 5F and I*), which is similar to what we observed with the wild-type RABX-5::GFP KI line. In contrast, when K323 is mutated, the signal from RABX-5(K323A)::GFP KI remains dim in both wild-type and *usp-50* mutant animals (*Figure 5G–I*). These observations suggest that USP-50 cannot regulate the endosomal localization of RABX-5 when K323 is mutated. Furthermore, the *rabx-5(K323R)* mutation successfully suppressed both the enlarged EE and diminished LE phenotypes of *usp-50* (*Figure 5J–M*). Taken together, USP-50 recruitment to EEs relies on RABX-5, and through its deubiquitination action on the Vps9 domain of RABX-5, USP-50 dissociates RABX-5 from EEs, thereby terminating Rab5 signaling to promote endosome maturation.

## USP-50/USP8 is required for SAND-1/Mon1 recruitment

SAND-1/Mon1-Ccz1 binds to RABX-5, and by displacing RABX-5 from the endosomal membrane, functions as a GEF of Rab7 to recruit and activate Rab7 GTPase (*Nordmann et al., 2010*; *Poteryaev et al., 2010*). In the absence of RABX-5 (*rabx-5 null*), the GFP::SAND-1 puncta are diminished (*Figure 6A, B, and D*) and the LAAT-1::GFP-labeled lysosome structures are also reduced (*Figure 6E and F* and *Figure 6—figure supplement 1A*). In *usp-50* mutants, the RABX-5 signal is enhanced, while the lysosome structures are reduced. Intriguingly, when we introduced the GFP::SAND-1 marker into *usp-50* mutants, we found that the punctate distribution of GFP::SAND-1 is lost (*Figure 6A, C, and D*). Mon1a and Mon1b are mammalian homologs of worm SAND-1. In SUM159 cells, mEGFP-tagged endogenous Mon1a (*Figure 6—figure supplement 1B*) is localized on both Rab5-positive EEs and Rab7-positive LEs (*Figure 6—figure supplement 1C and D*). In contrast, endogenous Mon1b (*Figure 6—figure supplement 1E*) is more localized on LEs (*Figure 6—figure supplement 1F and G*). When the expression of USP8 in the mEGFP-Mon1a$^{+/+}$ or mEGFP-Mon1b$^{+/+}$ cells was knocked down by siRNA, we found that the number of vesicles positive for mEGFP-Mon1a or mEGFP-Mon1b was greatly reduced (*Figure 6H and I*). Taken together, these results indicate that the function of USP8/USP-50 in endosomal localization of SAND-1/Mon1-Ccz1 is evolutionarily conserved. We noticed that when *sand-1* is mutated, the EEs are enlarged (*Figure 6J–L* and *Figure 6—figure supplement 1H*) and LEs/lysosomes become smaller (*Figure 6M–O* and *Figure 6—figure supplement 1I*), which is highly similar to *usp-50* mutants. Furthermore, Co-IP (co-immunoprecipitation) experiments indicated that the USP-50 protein is able to bind to SAND-1 (*Figure 6P*), which is consistent with the role of USP-50 in endosomal localization of SAND-1.

SAND-1/Mon1 is a GEF for Rab7, and therefore a reduced level of SAND-1/Mon1 may decrease the endosomal distribution of Rab7 (*Hiragi et al., 2022*; *Nordmann et al., 2010*; *Yong et al., 2023*). Indeed, with the GFP::RAB-7 KI line, we found that the punctate RAB-7 signal was greatly reduced by loss of function of *sand-1* (*Figure 6Q and R*). In *usp-50* mutants, the punctate GFP::RAB-7 KI signal is also reduced (*Figure 6S and T*). Given the coupled phenotype of enlarged EEs and smaller LEs/lysosomes in both *usp-50* and *sand-1* mutants, we wondered whether increasing Rab7 would enhance the endosome maturation process, thus overriding the EE enlargement defect in *usp-50* mutants.

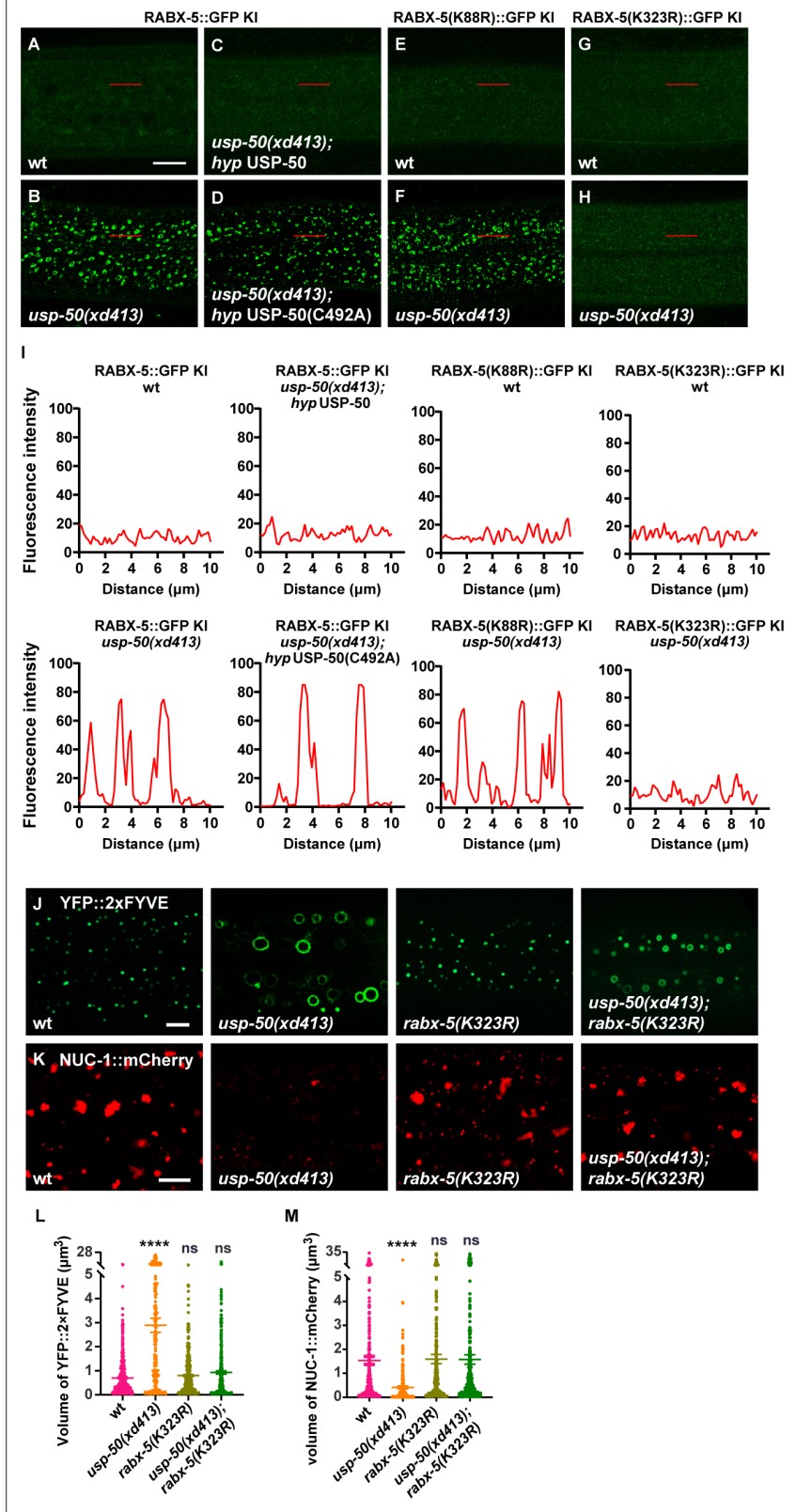

**Figure 5.** USP-50 dissociates RABX-5 from endosomes. (**A–D**) The punctate distribution of RABX-5::GFP KI (knock-in) in wild-type (**A**) and *usp-50(xd413)* (**B**). The increased RABX-5::GFP KI signal in *usp-50(xd413)* is rescued by expressing wild-type (**C**) but not C492A mutant *usp-50* (**D**). (**E, F**) The *usp-50(xd413)* mutation increases the GFP intensity of RABX-5(K88R)::GFP KI. (**G, H**) The *usp-50(xd413)* mutation does not alter the GFP intensity of RABX-

*Figure 5 continued on next page*

*Figure 5 continued*

5(K323R)::GFP KI. (**I**) Line scan analyses show the fluorescence intensity values along the red solid lines in (**A–H**). (**J, K**) Both the enlarged early endosome (EE) (YFP::2xFYVE) and the diminished late endosome (LE) (NUC-1::mCherry) phenotypes of *usp-50(xd413)* can be suppressed by *rabx-5(k323R)*. (**L, M**) Quantification of the volume of individual EEs (labeled by YFP::2xFYVE) and LEs (labeled by NUC-1::mCherry) in various genotypes. Data are presented as mean ± SEM. ****p<0.0001. ns, not significant; one-way ANOVA with Tukey's test. Scale bar represents 10 μm for (**A–H**), 5 μm for (**J, K**).

The online version of this article includes the following source data and figure supplement(s) for figure 5:

**Source data 1.** Excel file containing the quantified data of statistic analysis for *Figure 5I, L, and M*.

**Figure supplement 1.** USP8 inhibits the early endosome (EE) localization of Rabex5.

**Figure supplement 1—source data 1.** Original file for the western blot analysis in *Figure 5—figure supplement 1E* (anti-USP8 and anti-GAPDH).

**Figure supplement 1—source data 2.** PDF containing original scans of the relevant western blot analysis (anti-USP8 and anti-GAPDH) with highlighted bands and sample labels for *Figure 5—figure supplement 1E*.

**Figure supplement 1—source data 3.** Excel file containing the quantified data of statistic analysis for *Figure 5—figure supplement 1B and C*.

**Figure supplement 2.** Loss of *usp-50/USP8* leads to more activated RAB-5.

**Figure supplement 2—source data 1.** Original file for the Coomassie blue staining of purified GST-EEA1-NT protein in *Figure 5—figure supplement 2E*.

**Figure supplement 2—source data 2.** PDF containing the Coomassie blue staining of purified GST-EEA1-NT protein with highlighted bands and sample labels in *Figure 5—figure supplement 2E*.

**Figure supplement 2—source data 3.** Original file for the western blot analysis in *Figure 5—figure supplement 2F* (anti-GFP and anti-GST).

**Figure supplement 2—source data 4.** PDF containing original scans of the relevant western blot analysis (anti-GFP and anti-GST) with highlighted bands and sample labels for *Figure 5—figure supplement 2F*.

**Figure supplement 2—source data 5.** Original file for the western blot analysis in *Figure 5—figure supplement 2G* (anti-GFP and anti-Tub).

**Figure supplement 2—source data 6.** PDF containing original scans of the relevant western blot analysis (anti-GFP and anti-Tub) with highlighted bands and sample labels for *Figure 5—figure supplement 2G*.

**Figure supplement 2—source data 7.** Excel file containing the quantified data of statistic analysis for *Figure 5—figure supplement 2C, D, H, and J*.

**Figure supplement 3.** The ubiquitin modification sites on RABX-5.

Therefore, we overexpressed the wild-type *rab-7* gene and found that the EE enlargement phenotype of *usp-50* mutants was greatly suppressed (*Figure 6U–X*). In conclusion, we propose that the recruitment of USP-50/USP8, dependent on Rabex5, dissociates Rabex5 from EEs. Concurrently, this recruitment promotes the enrollment of SAND-1/Mon1-Ccz1, thereby facilitating the maturation of endosomes (*Figure 6Y*).

## Discussion

In this study, we identified that the recruitment of ubiquitin-specific protease USP-50/USP8 to EEs requires Rabex5. Instead of stabilizing Rabex5, the USP-50/USP8 recruitment dissociates Rabex5 from endosomes and meanwhile enrolls the Rab7 GEF SAND-1/Mon1. In *usp-50/usp8* loss-of-function cells, the RABX-5/Rabex5 signal is enhanced and the SAND-1/Mon1 protein fails to be localized onto endosomes. As a result, abnormal enlarged EEs are accumulated and the lysosomal structures become rudimentary.

Most studies of USP8 focus on endosomal trafficking of growth factor receptor tyrosine kinases (RTKs) in cultured vertebrate cells. In some cases, reduced USP8 activity results in accumulation of ubiquitinated cargoes (*Alwan and van Leeuwen, 2007*; *Bowers et al., 2006*; *Mizuno et al., 2006*; *Row et al., 2006*). USP8 can also promote RTK stability (*Berlin et al., 2010b*; *Mizuno et al., 2005*; *Niendorf et al., 2007*). It is thought that USP8 promotes the recycling of cell surface receptors back to the plasma membrane or enhances their degradation depending on when and where it deubiquitinates its substrate along the recycling pathway (*Mizuno et al., 2005*; *Niendorf et al., 2007*; *Wright*

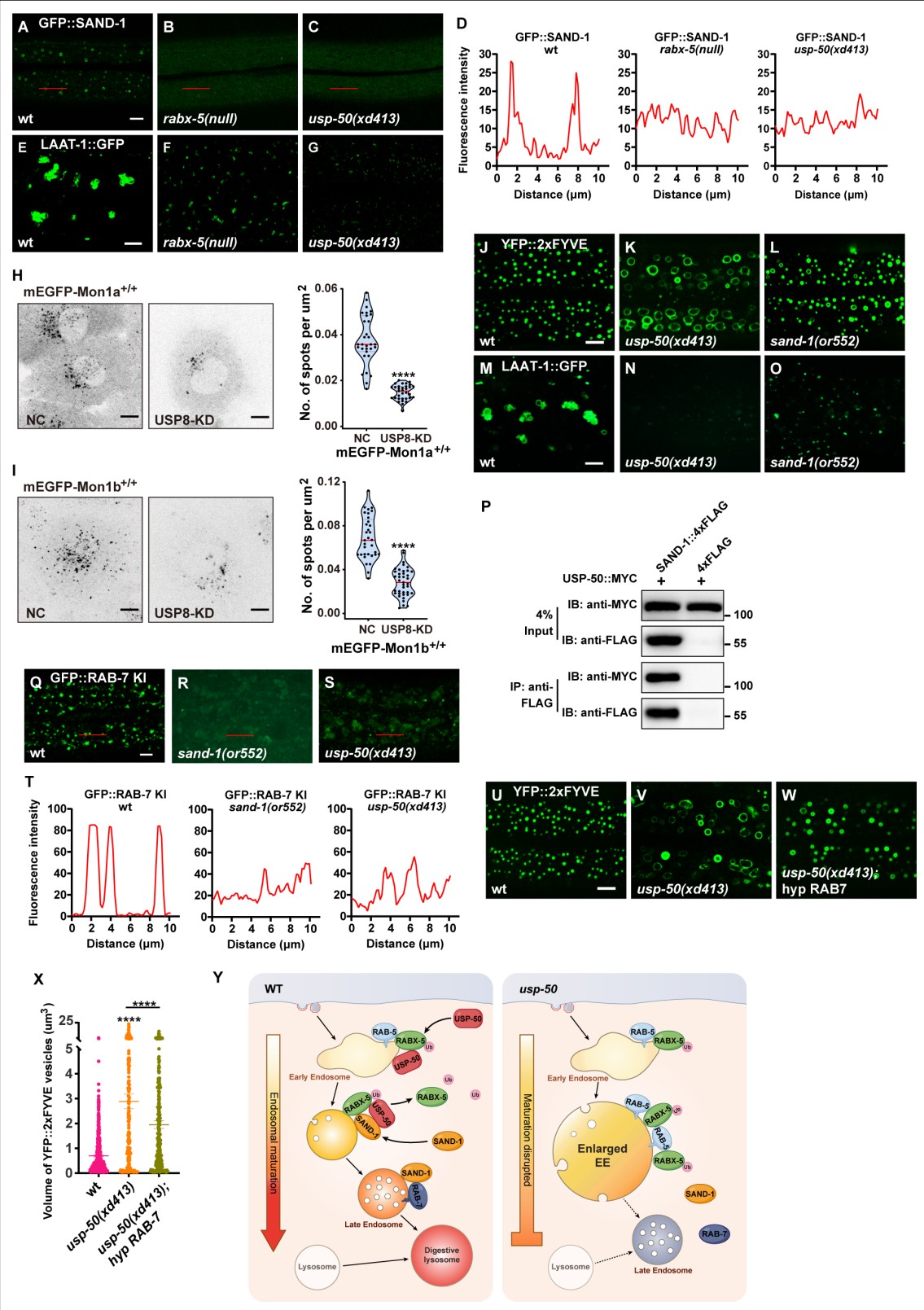

**Figure 6.** Loss of *usp-50/usp8* disrupts SAND-1/Mon1 localization. (**A–C**) The reduction of punctate GFP::SAND-1 signals in *rabx-5(null)* and *usp-50(xd413)* mutant animals. (**D**) Line scan analyses for (**A–C**). (**E–G**) The reduced lysosomes (labeled by LAAT-1::GFP) in *rabx-5(null)* and *usp-50(xd413)* mutant animals. (**H, I**) SUM159 cells genome-edited for mEGFP-Mon1a⁺/⁺ (**H**) or mEGFP-Mon1b⁺/⁺ (**I**) were treated with control siRNA or siRNA targeting USP8 and then imaged by spinning-disk confocal microscopy (starting from the bottom plane, spaced by 0.35 µm). The single-frame image shows the

*Figure 6 continued on next page*

*Figure 6 continued*

distribution and area of Mon1a- or Mon1b-labeled endosomes near the middle plane. The right panel shows quantification of the numbers of mEGFP-Mon1a$^{+/+}$ positive spots from 31 cells for wild-type and 33 cells for USP8-KD, and mEGFP-Mon1b$^{+/+}$ positive spots from 33 cells for wild-type and 39 cells for USP8-KD (****p<0.0001; unpaired Student's t-test.) Scale bar: 10 μm. (**J, L**) Early endosome (EEs) (labeled by YFP::2xFYVE) are enlarged in *sand-1(or552)* mutants. (**M–O**) Late endosomes/lysosomes (labeled by LAAT-1::GFP) are smaller in *sand-1(or552)* mutants. (**P**) USP-50::MYC and SAND-1::4xFLAG were expressed in HEK293T cells via co-transfection, then immunoprecipitated with anti-FLAG beads and immunoblotted with antibodies against MYC and FLAG. (**Q–S**) The reduction of punctate GFP::RAB-7 signals in *sand-1(or522)* and *usp-50(xd413)* animals. (**T**) Line scan analyses for (**Q–S**). (**U–W**) Overexpressing *rab-7* suppresses the enlarged EE phenotype of *usp-50* mutants. EEs were detected with YFP::2xFYVE. (**X**) Quantification of the individual volume of YFP::2xFYVE vesicles in hypodermal cell 7 (hyp7) of L4 stage from 10 animals for wild-type, 13 animals for *usp-50(xd413)*, and 10 animals for the *rab-7*-expressing line. Data are presented as mean ± SEM. ****p<0.0001; one-way ANOVA with Tukey's test. (**Y**) Working models showing the role of USP-50 in endosome maturation. RABX-5-depdendent recruitment of USP-50 promotes the dissociation of RABX-5 from EEs and enhances the recruitment of SAND-1, thereby promoting the endosome maturation process. Scale bar represents 5 μm for (**A–C, E–G, J–L, M–O, Q–S, U–W**), 10 μm for (**H, I**).

The online version of this article includes the following source data and figure supplement(s) for figure 6:

**Source data 1.** Original file for the western blot analysis in *Figure 6P* (anti-MYC and anti-FLAG).

**Source data 2.** PDF containing original scans of the relevant western blot analysis (anti-MYC and anti-FLAG) with highlighted bands and sample labels for *Figure 6P*.

**Source data 3.** Excel file containing the quantified data of statistic analysis for *Figure 6D, T, H, I, and X*.

**Figure supplement 1.** The distribution of endogenous Mon1a and Mon1b proteins.

**Figure supplement 1—source data 1.** Original file for the western blot analysis in *Figure 6—figure supplement 1B and E* (anti-mEGFP and anti-GAPDH).

**Figure supplement 1—source data 2.** PDF containing original scans of the relevant western blot analysis (anti-mEGFP and anti-GAPDH) with highlighted bands and sample labels for *Figure 6—figure supplement 1B and E*.

**Figure supplement 1—source data 3.** Excel file containing the quantified data of statistic analysis for *Figure 6—figure supplement 1A, H, and I*.

---

*et al., 2011*). Besides EGFR, USP8 regulates the endocytic trafficking and/or stability of many other transmembrane proteins (*Martín-Rodríguez et al., 2020*; *Peng et al., 2020*; *Sun et al., 2018*; *Xie et al., 2022*; *Xiong et al., 2022*), but conclusions about the impact of USP8 on protein stability are highly diverse. The conflicting results may be caused by massive global ubiquitination and proteolytic stress triggered by depletion of USP8 or overexpression of a catalytically inactive enzyme.

Endosome maturation controls the sorting, processing, recycling, and degradation of incoming substances and receptors, and is thus responsible for regulation and fine-tuning of numerous pathways in cells. The Rab5 GEF Rabex5 can be recruited to EEs through an early endosomal targeting domain or by binding with ubiquitinated cargoes through its UBD region (*Mattera and Bonifacino, 2008*; *Zhu et al., 2007*). Notably, complex intramolecular interactions are extensively involved in Rabex5 function and dynamic localization (*Lauer et al., 2019*). Here, RABX-5 associates with USP-50 through multiple domains. Thus, in the context of *usp8/usp-50* deletion, the enhanced endosomal localization of Rabex5/RABX-5 may be caused by alterations in multiple inter- and intramolecular interactions. The GTP-bound active Rab5 recruits Rabaptin-5 resulting in a positive feedback loop of Rab5 activation on endosomal membranes (*Zhang et al., 2014b*). How is this positive feedback loop terminated? The role of Rabex5 in recruiting Rab7 GEF SAND-1/Mon1-Ccz1 is well established. Here, we further showed that the endosomal localization of USP-50 is also dependent on RABX-5. Thus, Rabex5 may recruit multifaceted negative regulators, which work subsequently or collaboratively to regulate endosome maturation. LEs transport new lysosomal hydrolases and membrane proteins to lysosomes for the maintenance and amplification of the degradative compartment (*Yang and Wang, 2021*). Loss of worm *usp-50* results in reduced lysosome size. Previous studies also observed lysosome formation deficiency in fly *ubpy/usp8* knock-down fat cells (*Jacomin et al., 2015*; *Jacomin et al., 2016*). Removal of Rab5 and its replacement with Rab7 is an essential step in LE formation and in the transport of cargo to lysosomes (*Borchers et al., 2021*; *Zeigerer et al., 2012*). In the absence of USP8/USP-50, the RABX-5/Rabex5 signal is enhanced, but the endosomal localization of SAND-1/Mon1 is reduced, suggesting that in addition to Rabex5, USP8 is further needed to engage Rab7 GEF. The Mon1-Ccz1 complex can be recruited to various organelles through a variety of binding partners (*Gao et al., 2018*). Thus, recruited by Rabex5, USP8 may serve a linker specifically bridging endosomes to the Rab7 GEF. *sand-1* mutants display an almost identical phenotype to *usp-50* mutants, including enlarged EEs and much smaller LEs/lysosomes, implying that USP8/USP-50 functions

similarly to SAND-1/Mon1-Ccz1. In *usp-50* mutants, the great reduction of RAB-7 signal is accompanied by a dramatically increased RAB-5 signal. Therefore, we suspect that the extended Rab5 activation in *usp-50* mutants actually prevents the Rab5-to-Rab7 conversion from occurring. Of course, we cannot rule out the possibility that the remaining SAND-1 is able to convert some of the Rab5 to Rab7, thus forming LEs to some degree. However, due to the quick conversion of LEs to lysosomes for degradation, most of the EEs remain clearly differentiated from LEs/lysosomes (*Figure 2C and D* and *Figure 2—figure supplement 1A*). Overexpression of RAB-7 rescued the enlarged EE phenotype of *usp-50* mutants (*Figure 6U–X*), further supporting the idea that USP-50/USP8 downregulates Rab5 signaling and meanwhile promotes Rab7 activation, thus facilitating the EE-to-LE conversion. Together, we propose a working model, in which Rab5-coated vesicles recruit USP8, possibly through RABX-5-USP-50 interactions. Subsequently, USP8 dissociates Rabex5 from endosomes, meanwhile facilitating the recruitment of SAND-1/Mon1-Ccz1 complex to initiate LE formation (*Figure 6Y*).

Formation of ILVs is a hallmark of MVBs, which constitute morphologically distinct late endosomal structures that receive cargo in transit to the lysosomes. USP8 is important for the stability and ubiquitination status of various ESCRT components (*Adoro et al., 2017*; *Crespo-Yâñez et al., 2018*; *Mathieu et al., 2022*; *Mizuno et al., 2006*; *Niendorf et al., 2007*; *Zhang et al., 2014a*). Indeed, the punctate distribution of ESCRT-0 components is reduced significantly in *usp-50* worms (*Figure 1G and H* and *Figure 1—figure supplement 1B*). By EM ultrastructural analysis, we found that a large number of abnormal vesicular structures accumulate in *usp-50* mutants and a large portion of them contain various intraluminal structures (*Figure 2L–Q*). In wild-type animals, MVBs are rarely observed, indicating that once formed, they rapidly fuse with lysosomes and are subsequently degraded. The loss of USP-50 results in the upregulation of Rab5 signaling, which may enhance the homophilic fusion of EEs. Given the substantial reduction in Rab7 levels and the marked diminution in lysosomal structures, it is likely that the aberrantly enlarged vesicles in *usp-50* mutants represent excessively enlarged EEs.

USP8 deubiquitinates numerous plasma membrane receptors, making this enzyme a promising target in cancer therapy to overcome chemoresistance associated with RTK stabilization (*Byun et al., 2013*; *Islam et al., 2021*). Gain-of-function mutations of USP8 have been found in microadenomas of patients with Cushing's disease, a rare disease where the secretion of large amounts of adrenocorticotrophic hormone by pituitary corticotroph adenomas results in an excess of glucocorticoids and hypercortisolism, putatively due to defective EGFR sorting (*Ma et al., 2015*; *Reincke et al., 2015*). The role of USP8 in directing endosomal trafficking revealed here should shed new light on understanding its contribution to membrane receptor trafficking, resistance to chemotherapy, or EGFR stabilization in Cushing's disease.

# Materials and methods

## *C. elegans* genetics

Strain maintenance and genetic manipulations were performed as described (*Brenner, 1974*). The following strains were used in this study: linkage group (LG) I: *stam-1(ok406)*; LG III: *rabx-5(xd548)*; LG IV: *sand-1(or552)*; LG V: *usp-50(gk632973)*, *usp-50(xd413)*. Mutants and GFP knock-in strains for *rabx-5* are: *xdKi58* (*rabx-5::gfp* knock-in), *xd571* (*rabx-5(K88R)::gfp* knock-in), *xd572* (*rabx-5(K323R)::gfp* knock-in). Additional knock-in strains are: *xdKi22* (*gfp::rab-5* knock-in), *xdKi18* (*gfp::rab-7* knock-in). The reporter strains used in this work are listed as follows: *xdEx2766* (*Psemo-1*::GFP::RAB-5), *xdEx2857* (*Psemo-1*::mCherry::RAB-5), *xdEx2860* (*Psemo-1*::USP-50::GFP), *xdEx2863* (*Psemo-1*::mCherry::RAB-7), *xdEx2991* (*Psemo-1*::USP-50(C492A)::GFP), *xdEx2949* (*Psemo-1*::GFP::SAND-1), *qxIs354* (*Pced-1*::LAAT-1::GFP), *qxIs257* (*Pced-1*::NUC-1::nmCherry), *qxIs352* (*Pced-1*::LAAT-1::nmCherry), *qxIs439* (*Psemo-1*::GFP::TRAM), *qxEx3928* (*Psemo-1*::MANS::GFP), *yqIs75* (*Pvps-29*::VPS-29::GFP), *bIs46* (*Pvit-2*::GFP::RME-1), *opIs334* (*Pced-1*::YFP::2xFYVE), and *Phgrs-1*::HGRS-1::GFP. The CRISPR/Cas9-mediated genome editing strains were generated by Sunny Biotech and were verified with DNA sequencing. All strains were outcrossed with wild-type twice before use.

## *C. elegans* gene expression constructs

The complete *usp-50* cDNA was provided by Yuji Kohara (National Institute of Genetics, Japan). The full-length wild-type and C492A mutant *usp-50* cDNAs were cloned into dpSM vector. The P*usp-50*, P*semo-1*, P*myo-3*, and P*vha-6* promoters were cloned into dpSM upstream of the *usp-50* cDNA. The

human *usp8* cDNA was cloned into dpSM vector after the P*usp-50* promoter. To express USP-50, SAND-1, RAB-5, or RAB-7 in hyp7, the P*semo-1* promoter was cloned into dpSM vector, followed by GFP, mCherry, or corresponding cDNAs. The full-length and truncated forms of *rabx-5, sand-1,* or *usp-50* cDNAs were cloned into pcDNA3.0 with MYC tag, or pCS2 with 4xFLAG tag. The N-terminus of EEA1 was cloned into pGEX4T-1 vector. All constructs were confirmed by DNA sequencing.

## *C. elegans* imaging analysis

Images of LAAT-1::GFP with NUC-1::mCherry were collected from 3-day-old adults. Images of YFP::2x-FYVE with LAAT-1::mCherry were collected from L4 worms. Images were captured by spinning-disk microscopy (Observer Z1; Carl Zeiss). The co-localization analysis was performed with ImageJ (NIH) with Pearson's correlation coefficient. At least 10 worms were examined. Fluorescence images of worms expressing LAAT-1::GFP, NUC-1::mCherry at adult day 3 and YFP::2xFYVE, HGRS-1::GFP, GFP::RAB-5 at the L4 stage were captured by spinning-disk microscopy (Observer Z1, Carl Zeiss) in 10–20 z-series (0.3 µm/section). A 3D view was reconstituted from the serial optical sections and the number and volume of endosomes and lysosomes were measured using Volocity software (Perkin-Elmer). At least 10 worms were examined for every genotype at each given developmental stage. The volumes of YFP::2xFYVE-marked vesicles from 10 wild-type worms and 13 *usp-50(xd413)* worms were measured and presented in *Figure 2B*, *Figure 2—figure supplement 1B*, *Figure 4I*, *Figure 5L*, *Figure 6X*, and *Figure 6—figure supplement 1H*. The volumes of LAAT-1::GFP-marked vesicles from 19 wild-type worms and 16 *usp-50(xd413)* worms were measured and presented in *Figure 1D* and *Figure 6—figure supplement 1A and I*.

To quantify the fluorescence intensity of USP-50::GFP, RABX-5::GFP, GFP::SAND-1, and GFP::R-AB-7, images of strains for comparison were captured with a 63× objective. Mean fluorescence intensity was determined by ImageJ. For line scan analyses, fluorescence intensity values along the solid lines of a given image were extracted with ImageJ software and plotted using GraphPad Prism 8.

## *C. elegans* high-pressure freezing electron microscopy

Young adult worms were cryofixed on a Leica Microsystems HPM 100 and frozen in liquid nitrogen. After high-pressure freezing, the samples were washed four times in acetone and stained with 1% uranyl acetate for 1 hr. The infiltration was performed by increasing the concentration of SPI-PON 812. Samples were placed in fresh resin in an embedding mold and polymerized in 60°C oven for 3 days. Thin sections (60 nm) were produced with a diamond knife (Diatome) on an ultramicrotome (Ultracut UC7; Leica Microsystems). Sections were pictured with a JEM-1400 TEM (Hitachi HT7700), operating at 80 kV. Pictures were recorded on a Gatan832 4 k × 2.7 k CCD camera. The sizes of vesicles detected in EM were evaluated by ImageJ.

## *C. elegans* protein expression and protein-protein interactions

HEK293T cells were cultured at 37°C with 5% $CO_2$ in DMEM supplemented with 10% FBS (Hyclone). Transfections were performed with Lipofectamine 3000 (Invitrogen) according to the manufacturer's instructions. Cultured cells were harvested 24–48 hr after transfection. For immunoprecipitation, whole-cell extracts were collected 24–48 hr after transfection and lysed in RIPA buffer (1% [vol:vol] Triton X-100, 100 mM Tris-HCl, 50 mM EDTA, 150 mM NaCl, 1% deoxycholate, 0.1% SDS with protease inhibitor cocktail; Roche, 04693132001). Worms were lysed with 1% Nonidet P-40 worm lysis buffer (40 mM Tris-HCl, 150 mM NaCl, 1% NP-40) and homogenized with a Dounce homogenizer (Cheng-He Company, Zhuhai, China) on ice for 10 min. Both cell lysates and worm lysates were centrifuged at 12,000 rpm for 15 min at 4°C and the protein supernatants were incubated with anti-FLAG M2 affinity gel or GFP Nanoab-Agarose. After 8 hr of incubation, the agarose beads were washed with lysis buffer. For protein purification, the immunoprecipitated samples were eluted with 3xFLAG peptide or glycine (pH 2.5) neutralized with Tris buffer (pH 10.4). Immunoprecipitated samples or whole-cell lysates were resolved by SDS-PAGE, then transferred to nitrocellulose membranes (Pall Life Sciences). Membranes were blocked with 5% dried milk and signals were visualized with Pierce ECL western blotting substrate (Thermo Fisher Scientific, 34095).

## *C. elegans* GTP-RAB-5 pull-down assay

GST or the GST-EEA1-NT fusion protein was expressed in *Escherichia coli* BL21 (DE3) and induced with 0.2 mM IPTG for 12 hr at 25°C. Bacterial pellets were lysed by sonication in PBS buffer containing 1% Triton X-100, 1 mM phenylmethylsulfonyl fluoride, and protease inhibitor cocktail. GST or the GST fusion protein was purified using a glutathione Sepharose 4B column (GE Healthcare). GFP::RAB-5 protein from wild-type or *usp-50(xd413)* mutant GFP::RAB-5 KI worms was collected and washed with M9 buffer. Worm lysis buffer containing 1% Nonidet P-40 was then added and samples were disrupted with a Dounce homogenizer on ice for 10 min. Debris was removed by centrifuging at 12,000 rpm for 15 min at 4°C. The GFP::RAB-5 input in each experiment was equalized before the pull-down assay. The worm lysates were incubated with GST or GST-EEA1-NT coupled to glutathione Sepharose 4B for 4 hr at 4°C. After washing five times, the GFP::RAB-5 protein was analyzed by 12% SDS-PAGE followed by standard western blotting with anti-GFP antibody.

## *C. elegans* membrane/cytoplasm ratio of GFP::RAB-5

Images of GFP::RAB-5 KI worms were collected at the L4 stage using spinning-disk confocal microscopy (Observer Z1, Carl Zeiss). The total fluorescence intensity and membranous fluorescence intensity were measured using Volocity software (PerkinElmer). The cytosol fluorescence intensity was measured as total fluorescence intensity minus the membranous fluorescence intensity. Then the membrane/cytoplasm fluorescence intensity ratio was measured. At least 10 worms were examined for each genotype.

## *C. elegans* antibodies and reagents

The primary antibodies used were: anti-GFP (Abcam, ab290), anti-FLAG (Sigma-Aldrich, F1804), anti-tubulin (Sigma-Aldrich, T5168), anti-myc (Santa Cruz Biotechnology, sc-40). The secondary antibodies used were: goat anti-rabbit IgG-HRP (Santa Cruz Biotechnology, sc-2004) and goat anti-mouse IgG-HRP (Santa Cruz Biotechnology, sc-2005). Anti-FLAG M2 Agarose Affinity Gel (A2220) and 3xFLAG Peptide (F4799) were from Sigma-Aldrich. GFP-Nanoab-Agarose was from Lablead (GNA-50-1000) and Glutathione Sepharose 4B was from GE Healthcare (17075601).

## *C. elegans* statistical analysis

For each western blot, at least three replications were chosen for quantitative analysis with ImageJ following the published protocol (*Gallo-Oller et al., 2018*). All graphical data are presented as mean ± SEM. Two-tailed unpaired Student's t-tests were performed for comparison between two groups of samples. To compare multiple groups, one-way analysis of variance (ANOVA) followed by Tukey's post-test was performed. The p-values are represented as follows: *$p<0.05$, **$p<0.01$, ***$p<0.001$, ****$p<0.0001$, and NS (not significant, $p>0.05$).

## Cell culture

The human breast cancer SUM159 cells, kindly provided by J Brugge (Harvard Medical School, Boston, MA, USA), were confirmed by STR genotyping and verified to be mycoplasma-free using the TransDetect PCR Mycoplasma Detection Kit (TransGen Biotech). SUM159 cells were cultured at 37°C and 5% $CO_2$ in DMEM/F12 (Corning), supplemented with 5% FBS (Gibco), 100 U/ml penicillin and streptomycin (Corning), 1 µg/ml hydrocortisone (Sigma-Aldrich), 5 µg/ml insulin (Sigma-Aldrich), and 20 mM HEPES (Corning), pH 7.4.

## Plasmids and transfection

The cDNA sequence of human Rabex5 was amplified from SUM159 cDNA and inserted into a vector containing mEGFP to generate the plasmid Rabex5-mEGFP using the Gibson assembly method (pEASY-Uni Seamless Cloning and Assembly Kit, TransGen Biotech). The cDNA sequences of human Rab7a, Lamp1, EEA1, Rab5c-Q80L were amplified from the related cDNA clones and inserted into the mScarlet-I-, mEGFP-, or Halo-containing vectors to generate the plasmids mScarlet-I-Rab7a, Lamp1-mScarlet-I, mScarlet-I-EEA1, mScarlet-I-Rab5c, or Halo-Rab5c-Q80L using the Gibson assembly method. Transfections were performed using Lipofectamine 3000 Transfection Reagent (Invitrogen) according to the manufacturer's instructions. Cells expressing fluorescently tagged proteins at relatively low levels were imaged 16–20 hr after transfection.

## Genome editing of SUM159 cells to generate USP8-mEGFP[+/+], mEGFP-Mon1a[+/+], or mEGFP-Mon1b[+/+] cells using the CRISPR/Cas9 approach

SUM159 cells were genome-edited to incorporate mEGFP at the C-terminus of USP8, or N-terminus of Mon1a and Mon1b using the CRISPR/Cas9 approach as described (*He et al., 2017*; *Ran et al., 2013*). The single-guide RNAs (sgRNA) targeting human USP8 (5'-ATAACCTATGTCTCCTTATG-3'), human Mon1a (5'-GGATGGCTACTGACATGCAG-3'), or human Mon1b (5'-GATGTGCAGATGGAGG TCGG-3') were cloned into pSpCas9(BB)-2A-Puro (PX459) (Addgene #48139). The donor constructs used for homologous recombination were generated by cloning into the pUC19 vector with two ~600- to 800-nucleotide fragments of genomic DNA upstream and downstream of the stop codon of human USP8, or the start codon of Mon1a, and Mon1b, and the open reading frame of mEGFP using the pEASY-Uni Seamless Cloning and Assembly Kit (TransGen Biotech). A flexible (GGS)$_3$ linker was inserted between the start or stop codon of the gene and the open reading frame of mEGFP. 4–5 days after transfection with the donor plasmid and PX459 plasmid containing sgRNA targeting sequence, SUM159 cells expressing mEGFP were enriched by fluorescence-activated cell sorting (FACSAria Fusion, BD Biosciences). The sorted positive cells were expanded and then subjected to single-cell sorting into 96-well plates. The genome-edited monoclonal cell populations were identified by PCR (GoTaq Polymerase, Promega) and then verified by western blot analysis and imaging.

## Knockout of USP8 using the CRISPR/Cas9 approach

Knockout of USP8 in SUM159 cells was performed using the CRISPR/Cas9 approach as described (*He et al., 2017*; *Ran et al., 2013*). The sgRNA targeting human USP8 (5'-ATGCAGATTAGATCGTGATG -3') was cloned into pSpCas9(BB)-2A-Halo. pSpCas9(BB)-2A-Halo was generated by replacing GFP with HaloTag in pSpCas9(BB)-2A-GFP (PX458) (Addgene #488138). SUM159 cells were transfected with 1000 ng of the plasmid containing the sgRNA targeting sequence using Lipofectamine 3000. 24 hr after transfection, the cells expressing HaloTag were subjected to single-cell sorting into 96-well plates (FACSAria Fusion, BD Biosciences). The monoclonal cell populations with frameshift deletions in both alleles of *USP8* were identified by sequencing and confirmed by western blot analysis.

## Knockdown of USP8 using siRNA

The siRNA (GenePharma) used to knock down the expression of USP8 was transfected into cells by using Lipofectamine RNAiMAX (Invitrogen) according to the manufacturer's instructions. The siRNA sequence targeting human USP8 was 5'-CCAAAGAGAAAGGAGCAAT-3' (*Jing et al., 2020*). The non-targeting siRNA mixture (5'-ATGTATTGGCCTGTATTAG-3', 5'-GCGACGATCTGCCTAAGAT-3', and 5'-TTTCCGCACTGTGATTCGG-3') was used as a control. Knockdown of USP8 by siRNA was achieved by two sequential transfections on day 1 and day 3 (cells plated on day 0), followed by analysis on day 5.

## Live-cell imaging by spinning-disk confocal microscopy and imaging analysis

The spinning-disk confocal microscope was built on the Nikon TiE microscope as described (*Bi et al., 2021*). Briefly, the microscope was equipped with a CFI Apochromat SD 100× objective (1.46 NA, Nikon), a Motorized XY stage (Prior Scientific), a fully enclosed and environmentally controlled cage incubator (Okolab), OBIS 488, 561, and 647 nm lasers (Coherent), a CSU-X1 spinning-disk confocal unit (Yokogawa), and an EMCCD camera (iXon Ultra 897, Andor Technology). Images were acquired using Micro-Manager 2.0 (*Edelstein et al., 2010*).

SUM159 cells were plated on single-well confocal dishes (Cellvis) approximately 6–8 hr after transfection. Cells expressing relatively low levels of the endosome makers were imaged from the bottom surface (Z=1) to the middle plane (spaced 0.35 µm) in phenol-free DMEM/F12 (Corning) containing 5% FBS and 20 mM HEPES. Single frames or merge images were generated in Fiji (*Schindelin et al., 2012*). To quantify the areas and numbers of the fluorescently labeled Rab5c, Rabex5, EEA1, 2xFYVE, Mon1a, and Mon1b spots per cell, the cell boundary was first manually segmented based on the fluorescence of the cell in Fiji (*Schindelin et al., 2012*). The raw image and the segmented cell boundary were then loaded into Cellprofiler 4 (*Stirling et al., 2021*; http://www.cellprofiler.org) to detect the numbers and areas of spots in the middle plane of the cell (optical section Z=4). Finally, these results were exported from Cellprofiler 4 and further analyzed and plotted using GraphPad Prism 9.

For live-cell imaging and tracking of USP8 recruitment to Rab5-positive carriers, USP8-mEGFP[+/+] cells were transiently transfected with mScarlet-I-Rab5c, and then imaged at two planes (the bottom plane and the plane 0.5 μm above the bottom plane) every 2 s for 60 s by spinning-disk confocal microscopy. The maximum fluorescence intensity projection of the two imaging planes was generated using Fiji and was used for further imaging analysis. The detection and tracking of Rab5-positive carriers in the time-lapse series were performed using the TrackMate 7 plugin in Fiji (*Ershov et al., 2022*; *Tinevez et al., 2017*). The mScarlet-I-Rab5c channel was used for spot detection and tracking. Then the fluorescence intensities of mScarlet-I-Rab5c and USP8-mEGFP were extracted and plotted. Kymographs were generated using the Multi Kymograph plugin in Fiji.

### Imaging by SIM

Imaging of the sub-organelle distribution of USP8 on EEs was performed on the multi-SIM system as described (*Qiao et al., 2023*). The USP8-mEGFP[+/+] cells transiently expressed relatively low levels of mScarlet-I-EEA1 and the images were acquired with a CFI SR HP Apo TIRF 100× objective (1.49 NA, Nikon) and a sCMOS camera (Kinetix, TELEDYNE PHOTOMETRICS).

### Western blot analysis

SUM159 cells were lysed at 4°C for 20 min with RIPA lysis buffer (Sigma-Aldrich) containing a protease inhibitor cocktail (Thermo Scientific), and then pelleted at 12,000×$g$ for 15 min at 4°C. The supernatant was mixed with 5× sample buffer (MB01015, GenScript), heated to 100°C for 8–10 min, and then fractionated by SDS-PAGE (TGX FastCast AcrylamideKit, 10%, Bio-Rad) and transferred to nitrocellulose membranes (PALL). The membranes were incubated in TBST buffer containing 5% skim milk for 1 hr at room temperature, followed by overnight incubation at 4°C with the specific primary antibodies. After three washes in TBST (5 min each), the membranes were incubated with the appropriate HRP-conjugated secondary antibody (Beyotime, 1:1000) at room temperature for 1 hr. The membrane was incubated with the SignalFire ECL Reagent (Cell Signaling) or BeyoECL Moon (Beyotime) and imaged by the Tanon-5200 Chemiluminescent Imaging System (Tanon). The primary antibodies used in this study were: USP8 (sc-376130, 1:500, Santa Cruz), GFP (14-6674-82, 1:1000, Invitrogen), EEA1 (610456, 1:1000, BD Biosciences), GAPDH (60004-1-Ig, 1:50,000, Proteintech).

### Immunofluorescence

For immunofluorescence staining, cells were cultured on small coverslips (801010, NEST) coated with Poly-D-Lysine. Cells were washed with PBS once, and then fixed using 4% paraformaldehyde (157-8, Electron Microscopy Sciences) for 20 min at room temperature. After washing with PBS for three times, the samples were incubated with 0.5% Triton X-100 in PBS for 15 min at room temperature. Then the samples were incubated in the blocking buffer (1% BSA in PBS) for 60 min at room temperature. Afterward, the samples were incubated with the primary antibody against EEA1 (1:200 in 1% BSA) overnight at 4°C. After washing with PBS for five times, the samples were incubated with the Alexa Fluor 555-conjugated secondary antibodies (1:500 in 1% BSA, Thermo Fisher) for 60 min at room temperature. After washing with PBS for five times and ddH$_2$O once, the coverslip was mounted on a slide with the FluorSave Reagent (Merck, 345789-20ML). The prepared samples were imaged by the spinning-disk confocal microscope as described above.

## Acknowledgements

We thank Drs. Xiaochen Wang, Chonglin Yang, Qi Xie, Hong Zhang, Yuji Kohara, Xun Huang, Luke Lavis, Tom Kirchhausen, Joan Brugge, the Million Mutation Project, and the Caenorhabditis Genetics Center for providing reagents, strains, and technical support. This work was supported by the National Key R&D Program of China (2021YFA0805802 to MD and 2022YFA1304500 and 2021YFA0804802 to KH) and the National Natural Science Foundation of China (32070810 and 31921002 to MD and 32321004 and 92354305 to KH).

# Additional information

### Funding

| Funder | Grant reference number | Author |
| --- | --- | --- |
| National Key Research and Development Program of China | 2021YFA0805802 | Mei Ding |
| National Key Research and Development Program of China | 2022YFA1304500 | Kangmin He |
| National Key Research and Development Program of China | 2021YFA0804802 | Kangmin He |
| National Natural Science Foundation of China | 32070810 | Mei Ding |
| National Natural Science Foundation of China | 31921002 | Mei Ding |
| National Natural Science Foundation of China | 32321004 | Kangmin He |
| National Natural Science Foundation of China | 92354305 | Kangmin He |

The funders had no role in study design, data collection and interpretation, or the decision to submit the work for publication.

### Author contributions

Yue Miao, Conceptualization, Data curation, Formal analysis, Validation, Investigation, Visualization, Methodology, Writing – review and editing; Yongtao Du, Data curation, Formal analysis, Validation, Investigation, Visualization, Methodology, Writing – review and editing; Baolei Wang, Data curation, Formal analysis, Validation, Methodology; Jingjing Liang, Yu Liang, Song Dang, Methodology; Jiahao Liu, Visualization, Methodology; Dong Li, Resources, Supervision, Visualization, Methodology; Kangmin He, Conceptualization, Resources, Data curation, Formal analysis, Supervision, Funding acquisition, Investigation, Project administration, Writing – review and editing; Mei Ding, Conceptualization, Resources, Supervision, Funding acquisition, Investigation, Writing – original draft, Project administration, Writing – review and editing

### Author ORCIDs

Yue Miao ⓘ https://orcid.org/0000-0002-3317-0300
Yongtao Du ⓘ https://orcid.org/0009-0002-1340-9684
Kangmin He ⓘ https://orcid.org/0009-0004-5979-1617
Mei Ding ⓘ https://orcid.org/0000-0003-0576-6024

Reviewer #1 (Public review): https://doi.org/10.7554/eLife.96353.4.sa1
Reviewer #2 (Public review): https://doi.org/10.7554/eLife.96353.4.sa2
Reviewer #3 (Public review): https://doi.org/10.7554/eLife.96353.4.sa3
Author response https://doi.org/10.7554/eLife.96353.4.sa4

# Additional files

### Supplementary files
• MDAR checklist

### Data availability

All data generated or analysed during this study are included in the manuscript and supporting files; source data files have been provided for figures.

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
