## [Editor Report · eLife Assessment]

The manuscript presents an **important** model for the field of endosome maturation, providing perspective on the role of the deubiquitinating enzyme UPS-50/USP8 in the process. The evidence presented in the paper is clear, incorporating well-designed experiments that suggest the dual actions of UPS-50 and USP8 in the conversion of early endosomes into late endosomes. Overall, the work is **convincing** and centers on an intriguing subject.

---

## [Referee Report · Reviewer #1 (Public review)]

Summary:

The manuscript focuses on the role of the deubiquitinating enzyme UPS-50/USP8 in endosome maturation. The authors aimed to clarify how this enzyme drives the conversion of early endosomes into late endosomes. Overall, they did achieve their aims in shedding light on the precise mechanisms by which UPS-50/USP8 regulates endosome maturation. The results support their conclusions that UPS-50 acts by disassociating RABX-5 from early endosomes to deactivate RAB-5 and by recruiting SAND-1/Mon1 to activate RAB-7. This work is commendable and will have a significant impact on the field. The methods and data presented here will be useful to the community in advancing our understanding of endosome maturation and identifying potential therapeutic targets for diseases related to endosomal dysfunction. It is worth noting that further investigation is required to fully understand the complexities of endosome maturation. However, the findings presented in this manuscript provide a solid foundation for future studies.

Strengths:

The major strengths of this work lie in the well-designed experiments used to examine the effects of UPS-50 loss. The authors employed confocal imaging to obtain a picture of the aftermath of USP-50 loss. Their findings indicated enlarged early endosomes and MVB-like structures in cells deficient in USP-50/USP8.

Weaknesses:

Specifically, there is a need for further investigation to accurately characterize the anomalous structures detected in the ups-50 mutant. Also, the correlation between the presence of these abnormal structures and ESCRT-0 is yet to be addressed, and the current working model needs to be revised to prevent any confusion between enlarged early endosomes and MVBs.

---

## [Referee Report · Reviewer #2 (Public review)]

Summary:

In this study, the authors study how the deubiquitinase USP8 regulates endosome maturation in *C. elegans* and mammalian cells. The authors have isolated USP8 mutant alleles in *C. elegans* and used multiple in vivo reporter lines to demonstrate the impact of USP8 loss-of-function on endosome morphology and maturation. They show that in USP8 mutant cells, the early endosomes and MVB-like structures are enlarged while the late endosomes and lysosomal compartments are reduced. They elucidate that USP8 interacts with Rabx5, a guanine nucleotide exchange factor (GEF) for Rab5, and show that USP8 likely targets specific lysine residue of Rabx5 to dissociate it from early endosomes. They also find that localization of USP8 to early endosomes are disrupted in Rabx5 mutant cells. They observe that in both Rabx5 and USP8 mutant cells, the Rab7 GEF SAND-1 puncta which likely represents late endosomes are diminished, although that Rabex5 are accumulated in USP8 mutant cells. The authors provide evidence that USP8 regulates endosomal maturation in a similar fashion in mammalian cells. Based on their observations they propose that USP8 dissociates Rabex5 from early endosomes and enhances the recruitment of SAND-1 to promote endosome maturation.

Strengths:

The major highlights of this study include the direct visualization of endosome dynamics in a living multi-cellular organism, *C. elegans*. The high-quality images provide clear in vivo evidences to support the main conclusions. The authors have generated valuable resources to study mechanisms involved in endosome dynamics regulation in both the worm and mammalian cells, which would benefit many members in the cell biology community. The work identifies a fascinating link between USP8 and the Rab5 guanine nucleotide exchange factor Rabx5, which expands the targets and modes of action of USP8. The findings make a solid contribution toward the understanding of how endosomal trafficking is controlled.

Weaknesses:

- The authors utilized multiple fluorescent protein reporters, including those generated by themselves, to label endosomal vesicles. Although these are routine and powerful tools for studying endosomal trafficking, these results cannot tell that whether the endogenous proteins (Rab5, Rabex5, Rab7, etc.) are affected in the same fashion. Note that the authors have provided convincing evidence about the effects on Rab proteins in the revised manuscript.

- The authors clearly demonstrated a link between USP8 and Rabx5, and they showed that cells deficient of both factors displayed similar defects in late endosomes/lysosomes. But the authors didn't confirm whether and/or to which extent that USP8 regulates endosome maturation through Rabx5. Additional genetic and molecular evidence might be required to better support their working model. Note that the authors have provided convincing evidence about the role of USP8-Rabx5 axis in the revised manuscript.

---

## [Referee Report · Reviewer #3 (Public review)]

Summary:

The authors elucidated the role of USP8 in the endocytic pathway. Using *C. elegans* epithelial cells as a model, they observed that when USP8 function is lost, the cells have a decreased number and size in lysosomes. Since USP8 was already known to be a protein linked to ESCRT components, they looked into what role USP8 might play in connecting lysosomes and multivesicular bodies (MVB). They observed fewer ESCRT-associated vesicles but an increased number of abnormal enlarged vesicles (aberrant early endosomes) when USP8 function was lost. They showed that USP8 interacts with Rabx5 to dissociate it from early endosomes promoting the recruitment of the Rab7 GEF SAND-1/Mon1 and the maturation of the endosomes. The authors provided evidence that USP8 regulates endosomal maturation in a similar fashion in mammalian cells.

Strengths:

The use of two models, *C. elegans* and a mammalian cell line to describe a similar mechanism.

---

## [Author Response]

The following is the authors’ response to the previous reviews.

**Public Reviews:**

**Reviewer #1 (Public Review):**
**S**ummary:The manuscript focuses on the role of the deubiquitinating enzyme UPS-50/USP8 in endosome maturation. The authors aimed to clarify how this enzyme drives the conversion of early endosomes into late endosomes. Overall, they did achieve their aims in shedding light on the precise mechanisms by which UPS-50/USP8 regulates endosome maturation. The results support their conclusions that UPS-50 acts by disassociating RABX-5 from early endosomes to deactivate RAB-5 and by recruiting SAND-1/Mon1 to activate RAB-7. This work is commendable and will have a significant impact on the field. The methods and data presented here will be useful to the community in advancing our understanding of endosome maturation and identifying potential therapeutic targets for diseases related to endosomal dysfunction. It is worth noting that further investigation is required to fully understand the complexities of endosome maturation. However, the findings presented in this manuscript provide a solid foundation for future studies.

We thank this reviewer for the instructive suggestions and encouragement.

Strengths:The major strengths of this work lie in the well-designed experiments used to examine the effects of UPS-50 loss. The authors employed confocal imaging to obtain a picture of the aftermath of the USP-50 loss. Their findings indicated enlarged early endosomes and MVB-like structures in cells deficient in USP-50/USP8.

We thank this reviewer for the instructive suggestions and encouragement.

Weaknesses:Specifically, there is a need for further investigation to accurately characterize the anomalous structures detected in the *usp-50* mutant. Also, the correlation between the presence of these abnormal structures and ESCRT-0 is yet to be addressed, and the current working model needs to be revised to prevent any confusion between enlarged early endosomes and MVBs.

Excellent suggestions. USP8 has been identified as a protein associated with ESCRT components, which are crucial for endosomal membrane deformation and scission, leading to the formation of intraluminal vesicles (ILVs) within multivesicular bodies (MVBs). In *usp-50* mutants, we observed a significant reduction in the punctate signals of HGRS-1::GFP and STAM-1 (Figure 1G and H; and Figure1-figure supplement 1B), indicating a disruption in ESCRT-0 complex localization (Author response image 1). Additionally, lysosomal structures are markedly reduced in these mutants. In contrast, we found that early endosomes, as marked by FYVE, RAB-5, RABEX5, and EEA1, are significantly enlarged in *usp-50* mutants. Electron microscopy (EM) imaging further revealed an increase in large cellular vesicles containing various intraluminal structures. Given the reduction in lysosomal structures and the enlargement of early endosomes in *usp-50* mutants, these enlarged vesicles are likely aberrant early endosomes rather than late endosomal or lysosomal structures. To address potential confusion, we have revised the manuscript according to the reviewer's comments and updated the model to accurately reflect these observations.

**Reviewer #2 (Public Review):**
Summary:In this study, the authors study how the deubiquitinase USP8 regulates endosome maturation in *C. elegans* and mammalian cells. The authors have isolated USP8 mutant alleles in *C. elegans* and used multiple in vivo reporter lines to demonstrate the impact of USP8 loss-of-function on endosome morphology and maturation. They show that in USP8 mutant cells, the early endosomes and MVB-like structures are enlarged while the late endosomes and lysosomal compartments are reduced. They elucidate that USP8 interacts with Rabx5, a guanine nucleotide exchange factor (GEF) for Rab5, and show that USP8 likely targets specific lysine residue of Rabx5 to dissociate it from early endosomes. They also find that the localization of USP8 to early endosomes is disrupted in Rabx5 mutant cells. They observe that in both Rabx5 and USP8 mutant cells, the Rab7 GEF SAND-1 puncta which likely represents late endosomes are diminished, although Rabex5 is accumulated in USP8 mutant cells. The authors provide evidence that USP8 regulates endosomal maturation in a similar fashion in mammalian cells. Based on their observations they propose that USP8 dissociates Rabex5 from early endosomes and enhances the recruitment of SAND-1 to promote endosome maturation.

We thank this reviewer for the instructive suggestions and encouragement.

Strengths:The major highlights of this study include the direct visualization of endosome dynamics in a living multi-cellular organism, *C. elegans*. The high-quality images provide clear in vivo evidence to support the main conclusions. The authors have generated valuable resources to study mechanisms involved in endosome dynamics regulation in both the worm and mammalian cells, which would benefit many members of the cell biology community. The work identifies a fascinating link between USP8 and the Rab5 guanine nucleotide exchange factor Rabx5, which expands the targets and modes of action of USP8. The findings make a solid contribution toward the understanding of how endosomal trafficking is controlled.

We thank this reviewer for the instructive suggestions and encouragement.

Weaknesses:- The authors utilized multiple fluorescent protein reporters, including those generated by themselves, to label endosomal vesicles. Although these are routine and powerful tools for studying endosomal trafficking, these results cannot tell whether the endogenous proteins (Rab5, Rabex5, Rab7, etc.) are affected in the same fashion.

Good suggestion. Indeed, to test whether the endogenous proteins (Rab5, Rabex5, Rab7, etc.) are affected in the same fashion as fluorescent protein reporters, we supplemented our approach with the utilization of endogenous markers. These markers, including Rab5, RAB-5, Rabex5, RABX-5, and EEA1 for early endosomes, as well as RAB-7, Mon1a, and Mon1b for late endosomes, were instrumental in our investigations (refer to Figure 3, Figure 6, Figure 5-figure supplement 1, Figure 5-figure supplement 2, and Figure 6-figure supplement 1). Our comprehensive analysis, employing various methodologies such as tissue-specific fused proteins, CRISPR/Cas9 knock-in, and antibody staining, consistently highlights the critical role of USP8 in early-to-late endosome conversion.

- The authors clearly demonstrated a link between USP8 and Rabx5, and they showed that cells deficient in both factors displayed similar defects in late endosomes/lysosomes. However, the authors didn't confirm whether and/or to which extent USP8 regulates endosome maturation through Rabx5. Additional genetic and molecular evidence might be required to better support their working model.

Excellent point. To test whether USP-50 regulates endosome maturation through RABX-5, we performed additional genetic analyses. In *rabx-5(null)* mutant animals, the morphology of 2xFYVE-labeled early endosomes is comparable to that of wild-type controls (Figure 4H and I). Introducing the *rabx-5(null)* mutation into *usp-50(xd413)* backgrounds resulted in a significant suppression of the enlarged early endosome phenotype characteristic of *usp-50(xd413)* mutants (Figure 4H and I). These findings suggest that USP-50 may modulate the size of early endosomes through its interaction with RABX-5.

**Reviewer #3 (Public Review):**
Summary:The authors were trying to elucidate the role of USP8 in the endocytic pathway. Using *C. elegans* epithelial cells as a model, they observed that when USP8 function is lost, the cells have a decreased number and size in lysosomes. Since USP8 was already known to be a protein linked to ESCRT components, they looked into what role USP8 might play in connecting lysosomes and multivesicular bodies (MVB). They observed fewer ESCRT-associated vesicles but an increased number of "abnormal" enlarged vesicles when USP8 function was lost. At this specific point, it's not clear what the objective of the authors was. What would have been their hypothesis addressing whether the reduced lysosomal structures in USP8 (-) animals were linked to MVB formation? Then they observed that the abnormally enlarged vesicles, marked by the PI3P biosensor YFP-2xFYVE, are bigger but in the same number in USP8 (-) compared to wild-type animals, suggesting homotypic fusion. They confirmed this result by knocking down USP8 in a human cell line, and they observed enlarged vesicles marked by YFP-2xFYVE as well. At this point, there is quite an important issue. The use of YFP-2xFYVE to detect early endosomes requires the transfection of the cells, which has already been demonstrated to produce differences in the distribution, number, and size of PI3P-positive vesicles (doi.org/10.1080/15548627.2017.1341465). The enlarged vesicles marked by YFP-2xFYVE would not necessarily be due to the loss of UPS8. In any case, it appears relatively clear that USP8 localizes to early endosomes, and the authors claim that this localization is mediated by Rabex-5 (or Rabx-5). They finally propose that USP8 dissociates Rabx-5 from early endosomes facilitating endosome maturation.Weaknesses:The weaknesses of this study are, on one side, that the results are almost exclusively dependent on the overexpression of fusion proteins. While useful in the field, this strategy does not represent the optimal way to dissect a cell biology issue. On the other side, the way the authors construct the rationale for each approximation is somehow difficult to follow. Finally, the use of two models, *C. elegans* and a mammalian cell line, which would strengthen the observations, contributes to the difficulty in reading the manuscript.The findings are useful but do not clearly support the idea that USP8 mediates Rab5-Rab7 exchange and endosome maturation, In contrast, they appear to be incomplete and open new questions regarding the complexity of this process and the precise role of USP8 within it.

We thank this reviewer for the insightful comments. Fluorescence-fused proteins serve as potent tools for visualizing subcellular organelles both in vivo and in live settings. Specifically, in epidermal cells of worms, the tissue-specific expression of these fused proteins is indispensable for studying organelle dynamics within living organisms. This approach is necessitated by the inherent limitations of endogenously tagged proteins, whose fluorescence signals are often weak and unsuitable for live imaging or genetic screening purposes. Acknowledging concerns raised by the reviewer regarding potential alterations in organelle morphology due to overexpression of certain fused proteins, we supplemented our approach with the utilization of endogenous markers. These markers, including Rab5, RAB-5, Rabex5, RABX-5, and EEA1 for early endosomes, as well as RAB-7, Mon1a, and Mon1b for late endosomes, were instrumental in our investigations (refer to Figure 3, Figure 6, Figure 5-figure supplement 1, Figure 5-figure supplement 2, and Figure 6-figure supplement 1). Our comprehensive analysis, employing various methodologies such as tissue-specific fused proteins, CRISPR/Cas9 knock-in, and antibody staining, consistently highlights the critical role of USP8 in early-to-late endosome conversion. Specifically, we discovered that the recruitment of USP-50/USP8 to early endosomes is depending on Rabex5. However, instead of stabilizing Rabex5, the recruitment of USP-50/USP8 leads to its dissociation from endosomes, concomitantly facilitating the recruitment of the Rab7 GEF SAND-1/Mon1. In cells with loss-of-function mutations in *usp-50/usp8*, we observed enhanced RABX-5/Rabex5 signaling and mis-localization of SAND-1/Mon1 proteins from endosomes. Consequently, this disruption impairs endolysosomal trafficking, resulting in the accumulation of enlarged vesicles containing various intraluminal contents and rudimentary lysosomal structures.

Through an unbiased genetic screen, verified by cultured mammalian cell studies, we observed that loss-of-function mutations in *usp-50*/*usp8* result in diminished lysosome/late endosomes. Electron microscopy (EM) analysis indicated that *usp-50* mutation leads to abnormally enlarged vesicles containing various intraluminal structures in worm epidermal cells. USP8 is known to regulate the endocytic trafficking and stability of numerous transmembrane proteins. Given that lysosomes receive and degrade materials generated by endocytic pathways, we hypothesized that the abnormally enlarged vesicular structures observed in *usp-50* or *usp8* mutant cells correspond to the enlarged vesicles coated by early endosome markers. Indeed, in the absence of *usp8/usp-50*, the endosomal Rab5 signal is enhanced, while early endosomes are significantly enlarged. Given that Rab5 guanine nucleotide exchange factor (GEF), Rabex5, is essential for Rab5 activation, we further investigated its dynamics. Additional analyses conducted in both worm hypodermal cells and cultured mammalian cells revealed an increase of endosomal Rabex5 in response to *usp8/usp-50* loss-of-function. Live imaging studies further demonstrated active recruitment of USP8 to newly formed Rab5-positive vesicles, aligning spatiotemporally with Rabex5 regulation. Through systematic exploration of putative USP-50 binding partners on early endosomes, we identified its interaction with Rabex5. Comprehensive genetics and biochemistry experiments demonstrated that USP8 acts through K323 site de-ubiquitination to dissociate Rabex5 from early endosomes and promotes the recruitment of the Rab7 GEF SAND-1/Mon1. In summary, our study began with an unbiased genetic screen and subsequent examination of established theories, leading to the formulation of our own hypothesis. Through multifaceted approaches, we unveiled a novel function of USP8 in early-to-late endosome conversion.

**Recommendations for the authors:**

**Reviewer #1 (Recommendations For The Authors):**
(1) Within Figures 1K-N, diverse anomalous structures were detected in the *usp-50* mutant. Further scrutiny is needed to definitively characterize these structures, particularly as the images in Figures 1M and 1L exhibit notable similarities to lamellar bodies.

We thank the reviewer for the insightful question regarding the resemblance between the vesicles observed in our study and lamellar bodies (LBs). Lamellar bodies are specialized organelles involved in lipid storage and secretion1, prominently studied in keratinocytes of the skin and alveolar type II (ATII) epithelial cells in the lung2. These organelles contain not only lipids but also cell-type specific proteins and lytic enzymes. Due to their acidic pH and functional similarities, LBs are classified as lysosome-related organelles (LROs) or secretory lysosomes3,4. In *usp-50* mutants, we observed a considerable number of abnormal vesicles, some of which contain threadlike membrane structures and exhibit morphological similarities to LBs (Figure 2O). However, further analysis with a comprehensive panel of lysosome-related markers demonstrated a significant reduction in lysosomal structures within these mutants. In contrast, vesicles marked by early endosome markers, such as FYVE, RAB-5, RABX-5, and EEA1, were notably enlarged. These results suggest that the enlarged vesicles observed in *usp-50* mutants are more likely aberrant early endosomes rather than true lamellar bodies. We have revised the manuscript to reflect these findings and to clearly differentiate between these structures and lysosome-related organelles.

(2) The correlation between the presence of these abnormal structures and ESCRT-0 remains unaddressed, thus the assertion that UPS-50 regulates endolysosome trafficking in conjunction with ESCRT-0 lacks empirical support.

We thank the reviewer for the valuable suggestions. We apologize for any confusion and appreciate the opportunity to clarify our findings. The ESCRT machinery is essential for driving endosomal membrane deformation and scission, which leads to the formation of intraluminal vesicles (ILVs) within multivesicular bodies (MVBs). Recent research has shown that the absence of ESCRT components results in a reduction of ILVs in worm gut cells5. In wild type animals, the ESCRT-0 components HGRS-1 and STAM-1 display a distinct punctate distribution (Figure 1G and H). However, in *usp-50* mutants, the punctate signals of HGRS-1::GFP and STAM-1::GFP are significantly reduced (Figure 1G and H; and Figure 1-figure supplement 1B), indicating a role for USP-50 in stabilizing the ESCRT-0 complex. Our TEM analysis revealed an accumulation of abnormally enlarged vesicles containing intraluminal structures in *usp-50* mutants. When we examined a panel of early endosome and late endosome/lysosome markers, we found that early endosomes are significantly enlarged, while late endosomal/lysosomal structures are markedly reduced in these mutants. This suggests that the abnormal structures observed in *usp-50* mutants are likely enlarged early endosomes rather than classical MVBs. To further investigate whether the reduction in ESCRT components contributes to the late endosome/lysosome defects, we analyzed *stam-1* mutants. In these mutants, the size of RAB-7-coated vesicles was reduced (Author response image 1C), and the lysosomal marker LAAT-1 indicated a reduction in lysosomal structures (Author response image 1B). These results highlight the importance of the ESCRT complex in late endosome/lysosome formation. However, the morphology of early endosomes, as marked by 2xFYVE, remained similar to that of wild type in *stam-1* mutants (Author response image 1A). Therefore, while reduced ESCRT-0 components may contribute to the late endosome/lysosome defects observed in *usp-50* mutants, the enlargement of early endosomes in these mutants may involve additional mechanisms. We have revised the manuscript to incorporate these insights and to address the reviewer's comments more comprehensively.

**Author response image 1. sa4fig1:** (A) Confocal fluorescence images of hypodermis expressing YFP::2xFYVE to detect EEs in L4 stage animals in wild type and *stam-1(ok406)* mutants. Scale bar: 5 μm. (B) Confocal fluorescence images of hypodermal cell 7 (hyp7) expressing the LAAT-1::GFP marker to highlight lysosome structures in 3-day-old adult animals. Compared to wild type, LAAT-1::GFP signal is reduced in *stam-1(ok406)* animals. Scale bar: 5 μm. (C) The reduction of punctate endogenous GFP::RAB-7 signals in *stam-1(ok406)* animals. Scale bar: 10 μm.

(3) Endosomal dysfunction typically leads to significant alterations in the spatial arrangement of marker proteins across distinct endosomes. In the manuscript, the authors examined the distribution and morphology of early endosomes, multivesicular bodies (MVBs), late endosomes, and lysosomes in a *usp-50* deficient background primarily through single-channel confocal imaging. By employing two color images showing RAB-5 and RAB-7, in conjunction with HGRS-1, a more comprehensive picture of the aftermath of USP-50 loss can be obtained.

Good suggestions. We have conducted a double-labeling analysis to examine the distribution of RAB-5 and RAB-7 in conjunction with HGRS-1. In wild type animals, HGRS-1 exhibits a punctate distribution that is partially co-localized with both RAB-5 and RAB-7. In contrast, in *usp-50* mutants, the punctate signal of HGRS-1 is significantly reduced, along with its co-localization with RAB-5 and RAB-7 (Author response image 2). These results suggest that, in the absence of USP-50, the stabilization of ESCRT-0 components on endosomes is compromised.

**Author response image 2. sa4fig2:** ESCRT-0 is adjacent to both early endosomes and late endosomes. (A) Confocal fluorescence images of wild-type and *usp-50(xd413)* hypodermis at L4 stage co-expressing HGRS-1::GFP (*hgrs-1* promoter) and endogenous wrmScarlet::RAB-5. (B) HGRS-1 and RAB-5 puncta were analyzed to produce Manders overlap coefficient M1 (HGRS-1/RAB-5) and M2 (RAB-5/HGRS-1) (N=10). (C) Confocal fluorescence images of wild-type and *usp-50(xd413)* hypodermis at L4 stage co-expressing HGRS-1::GFP (*hgrs-1* promoter) and endogenous wrmScarlet::RAB-7. (D) HGRS-1 and RAB-7 puncta were analyzed to produce Manders overlap coefficient M1 (HGRS-1/RAB-7) and M2 (RAB-7/HGRS-1) (N=10). Scale bar: 10 μm for (A) and (C).

(4) The authors observed enlarged early endosomes in cells depleted of *usp-50*/usp8, along with enlarged MVB-like structures identified through TEM. The potential identity of these structures as the same organelle could be determined using CLEM.

We thank the reviewer for the valuable suggestion. Our TEM analysis identified a large number of abnormally enlarged vesicles with various intraluminal structures accumulated in *usp-50* mutants. As the reviewer correctly noted, CLEM (correlative light and electron microscopy) would be an ideal approach to further characterize these structures. We have been attempting to implement CLEM in *C. elegans* for a few years. Given that CLEM relies on fluorescence markers, in this study we focused on two tagged proteins, RAB-5 and RABX-5, which show enlargement in their vesicles in *usp-50* mutants. Unfortunately, we encountered significant challenges with this approach, as the GFP-tagged RAB-5 and RABX-5 signals did not survive the electron microscopy procedure. Attempts to align EM sections with residual GFP signaling yielded results that were not convincing. Consequently, we concentrated our analysis on a panel of molecular markers, including 2xFYVE, RAB-5, RABX-5, RAB-7, and LAAT-1. These markers consistently indicated that early endosomes are specifically enlarged in *usp-50* mutants, while late endosomal/lysosomal structures are notably reduced. Thus, the abnormal structures identified in *usp-50* mutants via TEM are likely to be enlarged early endosomes rather than the classical view of MVBs. We have revised the manuscript to reflect these findings and to clarify this point.

(5) The working model depicted in Figure 6 Y (right) requires revision, as it has the potential to mislead authors into mistaking enlarged early endosomes for multivesicular bodies (MVBs).

We thank the reviewer for the excellent suggestion. We have revised the model to clarify that it is the enlarged early endosomes, rather than MVBs, that are observed in *usp-50* mutants.

**Reviewer #2 (Recommendations For The Authors):**
(1) Is there any change of Rabx5 protein level in USP8/USP50 mutant cells?

Good question. In the absence of *usp-50/usp8*, we indeed observed a noticeable increase in the signal of Rabex5 on endosomes. To determine whether *usp-50/usp8* affects the protein level of Rabex5, we investigated the endogenous levels of RABX-5 using the RABX-5::GFP knock-in line. Compared to wild-type controls, we found an elevated protein level of RABX-5::GFP in the knock-in line (Author response image 3). This suggests that USP-50 may play a role in the destabilization of RABX-5/Rabex5 in vivo.

**Author response image 3. sa4fig3:** The endogenous RABX-5 protein level is increased in *usp-50* mutants. (A) The RABX-5::GFP KI protein level is increased in *usp-50(xd413)*. (B) Quantification of endogenous RABX-5::GFP protein level in wild type and *usp-50(xd413)* mutant animals.

(2) It is interesting that "The *rabx-5(null)* animals are healthy and fertile and do not display obvious morphological or behavioral defects.", which seems contrary to its role in regulating USP8 localization and endosome maturation.

It has been previously documented that *rabx-5* functions redundantly with *rme-6*, another RAB-5 GEF in *C. elegans*, to regulate RAB-5 localization in oocytes6. RNA interference (RNAi) targeting *rabx-5* in a *rme-6* mutant background results in synthetic lethality, whereas neither *rabx-5* nor *rme-6* single mutants are essential for worm viability. RME-6 co-localizes with clathrin-coated pits, while Rabex-5 is localized to early endosomes. Rabex-5 forms a stable complex with Rabaptin-5 and is part of a large EEA1-positive complex on early endosomes, whereas RME-6 does not interact with Rabaptin-5 (RABN-5) or EEA-1. These findings suggest that while RME-6 and RABX-5 may function redundantly, they likely play distinct roles in regulating intracellular trafficking processes. In the absence of RABX-5, USP-50 appears to lose its endosomal localization, although the size of the early endosome remains comparable to that of wild type. This observation contrasts with the phenotype associated with USP-50 loss-of-function, in which the early endosome is notably enlarged. These results suggest that residual USP-50 present in the endosomes is sufficient to maintain its role in the endocytic pathway. Conversely, the complete absence of USP-50 likely disrupts the transition of early endosomes to late endosomes, indicating a crucial role of USP-50 in this conversion process. It is also noteworthy that, although loss-of-function of *rabx-5* does not result in obvious changes to early endosomes, increasing the gene expression level of *rabx-5/Rabex-5* alone is sufficient to cause enlargement of early endosomes (Author response image 4) . Indeed, we observed that loss-of-function mutations in u_sp-50/usp_8 lead to abnormally enlarged early endosomes, accompanied by an enhanced signal of endosomal RABX-5. When the *rabx-5(null)* mutation was introduced into *usp-50* mutant animals, the enlarged early endosome phenotype seen in *usp-50* mutants was significantly suppressed (Figure 4H and I). This implies that maintaining a lower level of Rab5 GEF may be crucial for endolysosomal trafficking.

(3) Does Rabx5 mutation has any impact on early endosomes?

To address the question, we utilized the CRISPR/Cas9 technique to create a molecular null for *rabx*-5 (Figure 4E). In the *rabx-5(null)* mutant animals, we found that the 2xFYVE-labeled early endosomes are indistinguishable from wild type (Figure 4H and 4I). Given that r_abx-5_ functions redundantly with *rme-6*, another RAB-5 GEF in *C. elegans*, it is likely that the regulation of early endosome size involves a cooperative interaction between RABX-5 and RME-6.

(4) The authors observed a reduction of ESCRT-0 components in USP8 mutant cells, could this contribute to the late endosome/lysosome defects?

Good suggestion. In wild-type animals, the two ESCRT-0 components, HGRS-1 and STAM-1, exhibit a distinct punctate distribution (Figure 1G and H). However, in *usp-50* mutants, the punctate signals of HGRS-1::GFP and STAM-1::GFP are significantly diminished (Figure 1G and H; and Figure 1-figure supplement 1B), which aligns with the role of USP-50 in stabilizing the ESCRT-0 complex. To investigate whether the reduction in ESCRT components might contribute to defects in late endosome/lysosome formation, we examined *stam-1* mutants. In *stam-1* mutants, we observed a reduction in the size of RAB-7-coated vesicles (Author response image 1). Further, when we introduced the lysosomal marker LAAT-1::GFP into *stam-1* mutants, we found a substantial decrease in lysosomal structures compared to wild-type animals (Author response image 1). This suggests that the ESCRT complex is essential for proper late endosome/lysosome formation. In contrast, the morphology of early endosomes, as indicated by the 2xFYVE marker, appeared normal in *stam-1* mutants, similar to wild-type animals (Author response image 1). This implies that while a reduction in ESCRT-0 components may contribute to the late endosome/lysosome defects observed in *usp-50* mutants, the early endosome enlargement phenotype in _usp-5_0 mutants may involve additional mechanisms.

(5) Rabx5 is accumulated in USP8 mutant cells, I am very curious about the phenotype of USP8-Rabx5 double mutants. Could over-expression of Rabx5 (wild type or mutant forms) cause any defects?

Excellent suggestions. To address the question, we employed the CRISPR/Cas9 technique to create a molecular null for *rabx-5* (Figure 4E). In the *rabx-5(null)* mutant animals, we observed that the punctate USP-50::GFP signal became diffusely distributed (Figure 4F and G). This suggests that *rabx-5* is necessary for the endosomal localization of USP-50. Interestingly, in *rabx-5(null)* mutant animals, the 2xFYVE-labeled early endosomes appeared similar to those in wild-type animals (Figure 4H and I). When *rabx-5(null)* was introduced into *usp-50* mutant animals, the enlarged early endosome phenotype observed in *usp-50* was significantly suppressed (Figure 4H and I). This finding indicates that *usp-50* indeed functions through *rabx-5* to regulate early endosome size. Additionally, we constructed strains overexpressing either wild-type or K323R mutant RABX-5. Our results showed that overexpression of wild-type RABX-5 led to early endosome enlargement (as indicated by YFP::2xFYVE labeling) (Author response image 4A, B and D). In contrast, overexpression of the K323R mutant RABX-5 did not result in noticeable early endosome enlargement (Author response image 4A, C and D). Together, these data are in consistent with our model that USP-50 may regulate RABX-5 by deubiquitinating the K323 site.

**Author response image 4. sa4fig4:** (A-C) Over-expression wild type RABX-5 causes enlarged EEs (labeled by YFP::2xFYVE) while RABX-5(K323R) mutant form does not. (D) Quantification of the volume of individual YFP::2xFYVE vesicles. Data are presented as mean ± SEM. *****P*<0.0001. ns, not significant. One-way ANOVA with Tukey’s test.

(6) Rabx5 could be ubiquitinated at K88 and K323, and Rabx5-K323R showed different activity when compared with the wild-type protein in USP8 mutant cells. Could the authors provide evidence that USP8 could remove the ubiquitin modification from K323 in Rabx5 protein?

We appreciate the reviewer's insightful suggestions. To explore the potential of USP-50 in removing ubiquitin modifications from lysine 323 on the RABX-5 protein, we undertook a series of experiments. Initially, we sought to determine whether USP-50 influences the ubiquitination level of RABX-5 in vivo. However, due to the low expression levels of USP-50, we encountered challenges in obtaining adequate amounts of USP-50 protein from worm lysates. To overcome this, we expressed USP-50::4xFLAG in HEK293 cells for subsequent affinity purification. Concurrently, we utilized anti-GFP agarose beads to purify RABX-5::GFP from worms expressing the *rabx-5::gfp* construct. We then incubated RABX-5::GFP with USP-50::4xFLAG for varying durations and performed immunoblotting with an anti-ubiquitin antibody. As shown in Author response image 5A, our results revealed a decrease in the ubiquitination level of RABX-5 in the presence of USP-50, suggesting that USP-50 directly deubiquitinates RABX-5. Previous studies have indicated that only a minor fraction of recombinant RABX-5 undergoes ubiquitination in HeLa cells, which is believed to have functional significance7. Our findings are consistent with this observation, as only a small fraction of RABX-5 in worms is ubiquitinated. Rabex-5 is known to interact with both K63- and K48-linked poly-ubiquitin chains. To further elucidate whether USP-50 specifically targets K48 or K63-linked ubiquitination at the K323 site of RABX-5, we incubated various HA-tagged ubiquitin mutants with either wild-type or K323R mutant RABX-5 protein. Our results indicated that the K323R mutation reduces K63-linked ubiquitination of RABX-5 (Author response image 5). This experiment was repeated multiple times with consistent results. Additionally, while overexpression of wild-type RABX-5 led to an enlargement of early endosomes, as evidenced by YFP::2xFYVE labeling, overexpression of the K323R mutant did not produce a noticeable effect on endosome size (Author response image 4). Collectively, this finding indicates that RABX-5 is subject to ubiquitin modification in vivo and that USP-50 plays a significant role in regulating this modification at the K323 site.

**Author response image 5. sa4fig5:** (A) RABX-5::GFP protein was purified from worm lysates using anti-GFP antibody. FLAG-tagged USP-50 was purified from HEK293T cells using anti-FLAG antibody. Purified RABX-5::GFP was incubated with USP-50::4FLAG for indicated times (0, 15, 30, 60 mins), followed by immunoblotting using antibody against ubiquitin, FLAG or GFP. In the presence of USP-50::4xFLAG, the ubiquitination level of RABX-5::GFP is decreased. (B) Quantification of RABX-5::GFP ubiquitination level from three independent experiments. (C) HEK293T cells were transfected with HA-Ub or indicated mutants and 4xFLAG tagged RABX-5 or RABX-5 K323R mutant for 48h. The cells were subjected to pull down using the FLAG beads, followed by immunoblotting using antibody against HA or Flag.

(7) The authors described "the almost identical phenotype of usp-50/usp8 and sand-1/Mon1 mutants", found protein-protein interaction between USP8 and sand-1, and showed that sand1-GFP signal is diminished in USP8 mutant cells. These observations fit with the possibility that USP8 regulates the stability of sand-1 to promote endosomal maturation. Could this be tested and integrated into the current model?

are grateful for the insightful comments provided by the reviewer. Rab5, known to be activated by Rabex-5, plays a crucial role in the homotypic fusion of early endosomes. Rab5 effectors also include the Rab7 GEF SAND-1/Mon1–Ccz1 complex. Rab7 activation by SAND-1/Mon1-Ccz1 complex is essential for the biogenesis and positioning of late endosomes (LEs) and lysosomes, and for the fusion of endosomes and autophagosomes with lysosomes. The Mon1-Ccz1 complex is able to interact with Rabex5, causing dissociation of Rabex5 from the membrane, which probably terminates the positive feedback loop of Rab5 activation and then promotes the recruitment and activation of Rab7 on endosomes. In our study, we identified an interaction between USP-50 and the Rab5 GEF, RABX-5. In the absence of USP-50, we observed an increased endosomal localization of RABX-5 and the formation of abnormally enlarged early endosomes. This phenotype is reminiscent of that seen in *sand-1* loss-of-function mutants, which also exhibit enlarged early endosomes and a concomitant reduction in late endosomes/lysosomes. Notably, USP-50 also interacts with SAND-1, suggesting a potential role in regulating its localization. We could propose several models to elucidate how USP-50 might influence SAND-1 localization, including:

(1) USP-50 may stabilize SAND-1 through direct de-ubiquitination.

(2) In the absence of USP-50, the sustained presence of RABX-5 could lead to continuous Rab5 activation, which might hinder or delay the recruitment of SAND-1.

(3) USP-50 could facilitate SAND-1 recruitment by promoting the dissociation of RABX-5.

We are actively investigating these models in our laboratory. Due to space constraints, a more detailed exploration of how USP-50 regulates SAND-1 stability will be presented in a separate publication.

References:

(1) Schmitz, G., and Müller, G. (1991). Structure and function of lamellar bodies, lipid-protein complexes involved in storage and secretion of cellular lipids. J Lipid Res *32*, 1539-1570.

(2) Dietl, P., and Frick, M. (2021). Channels and Transporters of the Pulmonary Lamellar Body in Health and Disease. Cells-Basel *11*. https://doi.org/10.3390/cells11010045.

(3) Raposo, G., Marks, M.S., and Cutler, D.F. (2007). Lysosome-related organelles: driving post-Golgi compartments into specialisation. Current opinion in cell biology *19*, 394-401. https://doi.org/10.1016/j.ceb.2007.05.001.

(4) Weaver, T.E., Na, C.L., and Stahlman, M. (2002). Biogenesis of lamellar bodies, lysosome-related organelles involved in storage and secretion of pulmonary surfactant. Semin Cell Dev Biol *13*, 263-270. https://doi.org/10.1016/s1084952102000551.

(5) Ott, D.P., Desai, S., Solinger, J.A., Kaech, A., and Spang, A. (2024). Coordination between ESCRT function and Rab conversion during endosome maturation. bioRxiv, 2024.2005.2014.594104. https://doi.org/10.1101/2024.05.14.594104.

(6) Sato, M., Sato, K., Fonarev, P., Huang, C.J., Liou, W., and Grant, B.D. (2005). *Caenorhabditis elegans* RME-6 is a novel regulator of RAB-5 at the clathrin-coated pit. Nature cell biology *7*, 559-569. https://doi.org/10.1038/ncb1261.

(7) Mattera, R., Tsai, Y.C., Weissman, A.M., and Bonifacino, J.S. (2006). The Rab5 guanine nucleotide exchange factor Rabex-5 binds ubiquitin (Ub) and functions as a Ub ligase through an atypical Ub-interacting motif and a zinc finger domain. The Journal of biological chemistry *281*, 6874-6883. https://doi.org/10.1074/jbc.M509939200.